# A Comprehensive Review of Internet of Things: Technology Stack, Middlewares, and Fog/Edge Computing Interface

**DOI:** 10.3390/s22030995

**Published:** 2022-01-27

**Authors:** Omer Ali, Mohamad Khairi Ishak, Muhammad Kamran Liaquat Bhatti, Imran Khan, Ki-Il Kim

**Affiliations:** 1School of Electrical and Electronic Engineering, Universiti Sains Malaysia (USM), Nibong Tebal 14300, Malaysia; omerali@nfciet.edu.pk (O.A.); khairiishak@usm.my (M.K.I.); 2Department of Electrical Engineering, NFC Institute of Engineering & Technology (NFC IET), Multan 60000, Pakistan; dr_mklbhatti@nfciet.edu.pk; 3Department of Electrical Engineering, University of Engineering & Technology Peshawar, Peshawar 21500, Pakistan; imran_khan@uetpeshawar.edu.pk; 4Department of Computer Science and Engineering, Chungnam National University, Daejeon 34134, Korea

**Keywords:** Internet of Things (IoT), edge computing, fog computing, stack optimization, middleware, pervasive computing, ubiquitous computing

## Abstract

The Internet of Things (IoT) is an extensive network of heterogeneous devices that provides an array of innovative applications and services. IoT networks enable the integration of data and services to seamlessly interconnect the cyber and physical systems. However, the heterogeneity of devices, underlying technologies and lack of standardization pose critical challenges in this domain. On account of these challenges, this research article aims to provide a comprehensive overview of the enabling technologies and standards that build up the IoT technology stack. First, a layered architecture approach is presented where the state-of-the-art research and open challenges are discussed at every layer. Next, this research article focuses on the role of middleware platforms in IoT application development and integration. Furthermore, this article addresses the open challenges and provides comprehensive steps towards IoT stack optimization. Finally, the interfacing of Fog/Edge Networks to IoT technology stack is thoroughly investigated by discussing the current research and open challenges in this domain. The main scope of this study is to provide a comprehensive review into IoT technology (the horizontal fabric), the associated middleware and networks required to build future proof applications (the vertical markets).

## 1. Introduction

The Internet of Things (IoT) is widely regarded as one of the most prevailing technology revolutions of the previous two decades. IoT devices are often perceived as computing devices with sensing capabilities, onboard computational power, and an internet-enabled network to communicate with each other. It is one of the most rapidly growing areas that relies on machine-to-machine communications and utilizes an internet stack for end-to-end connectivity. In its simplest term, IoT is perceived as a network of billions of devices that can sense, actuate and relay the information to a centralized system. Nowadays, IoT devices and applications are deployed in various domains such as logistics, retail, health care, smart city network, intelligent transportation and disaster management. Despite the technological advancements in these individual domains, the heterogeneity of IoT devices and lack of standardization challenges are yet to be addressed. It is vital to examine the “Things” themselves, which operate differently depending on the implementation scenario, ranging from time-critical to mission-critical applications.

Despite the lack of standards, IoT systems are already expanding at a staggering pace. IoT systems have now exceeded the world’s population and are projected to hit an astounding 80 billion devices by 2025 [1,2]. In order to ensure the interoperability, scalability and reliability of IoT systems, a closer investigation of the technology stack is needed. This would make it easy to identify the technical issues that need to be resolved and standardized at each level [3]. One of the most significant challenges is accurately identifying billions of these devices in order to assure reliable data transmission between targeted devices. For such an expansion in current deployment rates and potential exponential growth projections for Internet-connected devices, a dynamic, scalable IP-addressing scheme is required [4,5]. Initially, internet addressing and routing relied on the IPv4 addressing protocol, which was deemed enough for identifying millions of devices worldwide. IPv4 addressing maintained unique and reliable addressing by incorporating a variety of addressing and translation technologies, including variable length subnet masking (VLSM), Classless Inter-domain Routing (CIDR), and Network address translation (NAT). However, the emerging trend of Internet-connected networks at large and consumer-specific IoT nodes, in particular, demands a holistic solution that is both flexible and intelligently addressable in order to satisfy the rising requirement for IP address space. The IP address scheme is considered to be the most important factor and the first step in the deployment and management of future IoT applications [6,7]. By 2025, it is anticipated that the total number of IoT device installations will exceed 75 billion globally. Researchers are already investigating the IPv6 addressing method for larger networks, which is projected to become the addressing standard for IoT devices.

The average roll-out pace and potential development forecast (as research reported from 2019 to 2030) both go hand in hand with the technological aspirations that will revolutionize the future interconnected world as shown in Figure 1. The most frequently used variables for forecasting the IoT growth trend are cost reductions in the device’s average operational and implementation costs. The productivity and commercial feasibility can only be improved by applying business intelligence and by combining existing standards with the proposed future IoT ecosystems [8], ensuring optimum resilience and adaptability [9,10,11]. IoT technologies enable the connectivity of real and virtual items, enabling a new era of ubiquitous computing. As a result, the Internet has transitioned from joining end-user nodes to interconnecting physical objects, resulting in a more intelligent object network capable of insightful collaboration and intelligent processing.

Nowadays, smart objects, Internet of Things (IoT) devices, and even Wireless Sensor Networks (WSN) are equipped with Ambient Intelligence (AmI), which integrates pervasive computing at the technological stack’s core. Recent studies over the last decades show an increase in smart IoT infrastructure deployments with elements of pervasive computing such as AmI, enabling business intelligence and decision-making capabilities [12]. Some of the technologies that are capable of introducing AmI to processes includes, Pervasive Computing (PC), Mobile Computing (MC) and Cyber-Physical Systems (CPS) [13]. These technologies are reliant on their hardware resources, node computing capability, critical security models, machine learning capabilities, and central processing. AmI provisioning in the technology stack requires the characterization of resources and business objectives at every layer of the technology stack.

To understand the interconnectedness of billions of these devices in pervasive and genuinely ubiquitous environments, it is necessary to explore real-world implementation and application scenarios. Smart cities are a great example of IoT devices working together towards AmI. Smart parking, remote monitoring, sidewalk monitoring, and real-time air quality monitoring are just a few examples of IoT sensors integrated in our ecosystems. Similarly, applications such as smart wearables have become an integral part of future health-care systems. Industrial IoT deployments have also grown significantly over the last decade. Integrating smart sensors improves the safety and security of industrial processes. To create a user-centric Internet of Things that connects people and their gadgets in a sustainable ecosystem, we must first address the integration of diverse technologies, the establishment of trusted communications, the management of huge amounts of data and services, and user engagement.

This article decouples the enabling technologies, the underlying infrastructure, and vendor-specific and vendor-neutral implementations of the IoT environment by contrasting existing tools on the Internet with the proposed IoT stack. The purposes and objectives of the present IoT implementations, as well as the need for standardisation, are described in further detail at each layer. Additionally, the role and behaviour of these devices that contribute to the creation of an ecosystem in terms of processing capability, resource availability, energy consumption, and sensing capabilities are also examined in detail. Nowadays, technological advancements are reshaping the industry toward small-form-factor, low-power gadgets that integrate a plethora of components. As a result, device cost, processing power, energy consumption, and interoperability with other systems are viewed as critical aspects in IoT applications [14,15,16,17].

Numerous studies have been conducted across multiple industry sectors on IoT devices, their technological stack, and application areas. The majority of the literature provides a high-level overview of efficacy and application in this domain. Additionally, there is a dearth of literature discussing IoT optimization and integration in the context of industry-specific deployments. However, combining this data and offering clear recommendations for IoT optimization and integration with future networks, such as edge computing, 5G networks, and beyond, remains a challenge, which served as the motivation for this study.

Some of the major study contributions can be given as follows.
A comprehensive insight into IoT technology stack, adaptation and growth trends.The detailed investigation of IoT Functional blocks at every layer (referred to as horizontal fabric), state-of-the-art research corresponding these elements, and the associated challenges.The characterization of Middleware, enterprise platforms and integration challenges for enterprise solutions (referred to as vertical markets).Future directions to optimize the IoT Technology Stack and its integration with enterprise systems.Interfacing Fog/Edge network to extend coverage, convergence and deployment scope for IoT networks.State-of-the-art research in Fog/Edge networks, open challenges and directions towards IoT interfacing, thus enhancing application of vertical markets.

Section 3 offers a comprehensive look into the industry outlook on the horizontal fabric (technology stack) and the vertical market (applications and services). Section 4 introduces a comprehensive layered IoT architecture that decouples devices, associated technologies, and standards on each layer. Centered on this, the entire Internet of Things (IoT) network topology is studied, focusing on the various layers that make up the horizontal IoT technology fabric. Section 5 furthers the fundamental building block of IoT networks by exploring block-level technologies in various layers of the current IoT system. It also investigates state-of-the-art technologies, standardisation initiatives, and research on the different layers of the system. Furthermore, the technology stack is examined in depth by decoupling the fundamental blocks of the IoT technology fabric.

Section 6 summarises the need for IoT network middleware and platforms by identifying device and technology specifications. In addition, new vertical market middleware technologies are being investigated for their effectiveness and security models. Finally, a range of analysis models focused on security, privacy, and trust were examined, and a vertical market gap was identified. Section 7 presents users with unique perspectives to unfold IoT topology, architecture, and fundamental blocks by addressing the entire IoT stack optimization opportunities.

Section 8 introduces the technical advances and open research problems in Fog/Edge computing as a major way forward in this article. A potential reference multi-tiered architecture model will also be provided to consider the significant components that will replace or apply to the current IoT architecture and deployments. A comprehensive survey covering the state-of-the-art studies into the underlying blocks of these technological principles was presented. Research challenges and forthcoming trends in this domain are also reported to provide a way forward for research in this domain.

## 2. Research Design

The primary research goals of this article are threefold. This article attempts to highlight the rapid growth in IoT technology, its adoption, application domains, and business opportunities. Next, the IoT technology stack, middleware integration, and the need for stack optimization are deeply investigated. Finally, the interfacing of Fog/Edge computing and associated technologies is investigated by looking into the state-of-the-art research and open challenges in this domain. In this regard, a rigorous systematic literature review (SLR) was conducted. This enabled us to examine potential future research directions in IoT technology and its associated application domains.

### Research Questions

By surveying the IoT technology stack from a layered perspective, we investigated the current technologies, growing trends, and emerging application domains. We found a slew of surveys and research papers focusing on the IoT technology stack, integration with existing systems, and highly specialised application domains like the Industrial Internet of Things (IIoT), IoT for wearable devices, renewable applications, and smart cities, among others. The role of middleware in current IoT solutions was also thoroughly examined. The impact of middlewares on the technology stack allowed for a deeper dive into IoT stack optimization, which is one of the article’s unique features. Finally, the interoperability of existing IoT solutions with fog/edge architectures and associated technologies like 5G was thoroughly investigated. We discovered a small number of surveys that specifically and individually reported these aspects, which served as the impetus for this study.

In addition, we investigated leading research and survey papers on IoT, its application scenarios, implementation details, and the open challenges in the technology stack that are impeding its growth. Although most researchers concentrated on general IoT application domains, as shown in Table 1. To the best of our knowledge, no previous research explicitly aligned the IoT technology stack and its interaction with Fog/Edge networks by offering unambiguous optimization recommendations.

In addition, we developed research questions based on those proposed by [33,34,35] to find research gaps in these domains. This prompted the formulation of (RQ1): “What is the current state of IoT technology stack (referred to as horizontal fabric) and application scenario (referred to as vertical markets)?”. This motivated us to expand our research into (RQ2): “What is the impact of utilizing middlewares into existing enterprise IoT applications?”. This naturally led to (RQ3): “What are the current technological and integration challenges, and how can the current technology stack be optimized?”. Finally, to enhance the coverage and application efficacy of current IoT networks, (RQ4): “How can Fog/Edge networks extend the capabilities of current IoT applications?”.

As a result, this review article is organized around the following research questions:What is the current state of IoT technology stack (referred to as horizontal fabric) and application scenario (referred to as vertical markets)? This question aims to identify the current state-of-the-art of IoT technology, growth trends, associated challenges and the range of applications and domains.What is the impact of utilizing middlewares in existing enterprise IoT applications? This question allowed us to classify the current state of middlewares currently being deployed for enterprise applications.What are the current technological and integration challenges, and how can the current technology stack be optimized? This question focuses on the integration effect, feasibility, and scope of these IoT application domains. It further aims at providing gaps and solutions to optimize the IoT technology stack from a layered perspective.How can Fog/Edge networks extend the capabilities of current IoT applications? This question is aimed at investigating the current state of Fog/Edge networks and the possibilities of extending these services to IoT deployments.

## 3. IoT Market Growth by Industry Sectors

According to reports, IoT sales and spending have increased dramatically in recent years. Businesses are eager to invest in technologies that generate user data for use in business analytics and decision-making [10]. IoT spending has increased by more than 200 percent in yearly expenditures in the domains of logistics and transportation alone, with business investments of up to 40 billion dollars. The market growth trends in various IoT domains are presented in Figure 2, where annual spending is studied over the course of last five years. In terms of market trends, revenues invested, and projected IoT rollout estimates, it is critical to analyse the relationship between industrial needs and technical resource availability, which will define this smart connected environment [36].

As reported earlier, the number of IoT device install-bases is expected to reach 75 billion by 2025 [1]. In the same perspective, smart objects (internet-connected devices with minimal computing capabilities) are expected to reach almost 200 billion entities by the end of 2021 [37,38]. One such example of a smart object is Amazon’s dash button that enables a pre-programmed purchase of goods with a press of a button [39]. The volume of data flows or traffic in general is another important indicator that highlights industry growth and commercial potential. At present, the global market is data-centric, and IoT systems are projected to generate data in magnitudes ranging from exabytes to zettabytes each year [37]. The global increase in M2M traffic alone is projected to increase by 51% by 2022. The Cisco forecast on these smart objects’ adaptation rate projects an increase of 17 percent in the compound annual growth rate of these devices [9]. The forecast on such smart devices presents a considerable increase in M2M traffic, as shown in Figure 3. Many IoT-based eco-systems, ranging from manufacturing, transportation, defence, construction, and waste management to the health care industry, witnessed significant increases in both adaption and financial advantage.

The global market prediction for IoT-based technologies and industries is crucial in the information and communications technology (ICT) sector, where global market values are estimated to reach USD 1102.6 billion by 2026 [40,41]. Another important aspect is the predicted global technology investment in IoT products, which is expected to reach USD 1.2 trillion by the end of current decade [11].

On the other hand, few researchers are looking to quantify IoT in novel ways by pooling the total number of M2M connections in capital growth. One such research by Frontier Economics forecasts a model that utilizes the projected data for the years 2018 to 2032. According to their model, a meager 10% growth in M2M connectivity could increase GDP by $2.26 trillion for the US alone [42]. Another approach to look at IoT industry patterns is through technology adaptation and Return on Investments (ROI). With the new industrial revolution, every major participant in the industry will be forced to modify or adapt to IoT-based infrastructure. However, from a research point of view, it is important to remember how the consumer compares the value generated toward ROI by implementing these devices or systems of interconnected devices [43].

According to Accenture Analytics, the true value of smart technology can only be realised through a thorough examination of big data from these embedded devices when exchanged with enterprise organisations. The research based on enterprise case study demonstrates that about 73% of the businesses had already started to implement over 20% of their technical budget in Big Data analytics [44]. This illustrates a rare chance that data mining, artificial learning, and business analytics have become key topics, mostly in the field of IoT. However, several businesses consider the security and privacy risks associated with IoT-generated data to be critical. Enterprises believe that device heterogeneity and limited on-board resources are the fundamental reasons that make IoT data vulnerable to theft and cyber attacks at almost every layer [45]. While the IoT domain has not yet been standardised, the severe privacy and governance standards on user data that govern the whole data-centric business remain in place. Furthermore, privacy laws and data security policies vary by area, making it nearly impossible for enterprises to compete in IoT and anticipate comparable ROIs [46].

This section detailed the market dynamics, development, and commercial opportunities in the rapidly increasing field of IoT. The numbers accurately depict the progress of these intelligent devices in terms of deployment and adaption. The rise in M2M traffic is predicted to result in a significant increase in both mobile and ISP traffic. Businesses examine scenarios in order to solve existing problems and develop a comprehensive business strategy that significantly increases their corporate position through capital development and investment returns. However, a sizable proportion of enterprises and consumers are concerned about the security of IoT data, presenting additional challenges to be addressed.

## 4. IoT Architectures, Platforms and Technology Stack

The modern Internet is a complex blend of Internet nodes, IoT devices, and smart objects. Internet-enabled networks require the implementation of an IP stack for communication between networks of objects. With the expansion of Internet networks, enterprises as well as research communities are investing in flexible and scalable IP networks for the future [13]. Currently, the Internet-enabled networks vary greatly both in technical implementations and in end-application needs. A conventional computing node (such as Personal Computers, Laptops, Mobiles and Tablets) implements an entire TCP/IP stack based on the Open Systems Interconnection (OSI) Model. However, due to the limited resources available on IoT devices, a lightweight IP stack is normally implemented. The on-board resource availability and energy consumption of the device primarily regulate the implementation of suitable protocols and standards in IoT devices. Therefore, it is fundamentally important to investigate the IoT architectures and platforms to understand the role and behavior at every layer of the technology stack.

In this review, we investigate the heterogeneity of the IoT fabric, from physical to the application layers, and refer to them as horizontal fabric [47]. On the one hand, researchers are attempting to integrate AmI into the IoT technological core, which is expected to improve business intelligence. In practise, however, IoT devices are primarily confined to sensing and relaying data. The restricted availability of on-board resources (such as processing, networking, and storage) directly contributes to these limitations, leaving the heavy lifting to the cloud. At the same time, integration of middlewares and edge networks now supports an increase in consumer-specific application developments in the vertical markets [48,49]. As a result, it is critical to investigate the technology stack (horizontal fabric) in relation to the application goals (the vertical markets).

The horizontal fabric is made up of “Things” and the communication stack, whereas middlewares, edge networks, and the cloud make application development simpler and thus enable vertical markets on top of this fabric [50]. A closer look at Figure 4 reveals a three-tier architectural model for IoT systems. A subtle balance between components and processes is presented in this model, where devices or “Things” are the primary layer of the architecture. It is apparent that this layer is open for vendor-specific implementations, resulting in a myriad of distinct components, modules and operating systems.

This layer is responsible for translating and propagating the heterogeneity of the entire IoT stack. The network layer exhibits similar variation in terms of component, module, and operating system selection. Devices are connected to the cloud at this layer, either through gateway devices or through a fog/edge network interface. The cloud layer is in charge of handling raw data from billions of IoT devices. To integrate business intelligence, current cloud-driven corporate systems offer solutions based on Artificial Intelligence (AI) and Machine Learning (ML). The cloud layer is dominated mostly by enterprise solutions that offer a variety of IoT network applications. While each cloud technology is different, the heterogeneity of this layer is often defined by the efficacy of the application design.

Since the Internet of Things primarily relies on the internet to connect these devices to services, most reference models and architectures use a layered approach to understanding and defining the functions at each layer. [47]. The alignment of this reference architecture model in layers that greatly simplify overall design goals is a popular trend. However, it is important to remember that most of these reference models do not correspond to the Internet or the TCP/IP stack [51]. It is indeed worth noting that the majority of alliances and organisations contribute to the standardisation of IoT protocol and architecture stacks by proposing their own reference models. This section looks into several of these models, which range from three-layer to middleware and five-layer models, as given in the literature [12,52,53].

Most architecture models now support the inclusion of middleware as a software-based interface between IoT processes and components. Middleware enables reliable and efficient communication between elements that are mostly not supported within the native operating systems. This results in simpler and standardised communication between processes, components, and devices, thus enabling a software interface that can extend to upper layers and promote vertical market integration. Cruz et al. [54] discussed one such middleware reference model in-depth, thoroughly presenting the current implementations and future perspectives of adopting one such standardised model. The researchers argued that the component level heterogeneity can only be mitigated by a standardised interface between processes that can be achieved through middleware platforms. In addition, the difficulties in standardisation were deeply investigated, and a middleware-based architecture model was proposed as a reference mode. Spies et al. [55] presented a different perspective on middleware technologies and challenges while focusing on a Service Oriented Architecture (SOA) based model, where vertical integration seems possible with cost effectiveness. The SOA-based middleware architecture presents layers as web services that can be scaled and concurrently inherited [56,57].

uBiuitous, secUre inTernet-of-things with Location and contExt-awaReness (BUTLER) is one of the first European consortium research projects to focus on pervasive computing and security for IoT applications in various domains. One of the project’s main objectives was to create context-aware and secure apps for a variety of deployment scenarios (such as healthcare, transportation, smart offices, and smart homes). The researchers proposed a device-centric architecture that included smartObjects, smartMobiles, and smartServers as three key components. The BUTLER project examines technology versus integration concerns using a five-layer architecture that closely resembles the internet layer model, with enabling technologies that help develop the horizontal fabric and address vertical integration problems [58] as shown in Figure 5.

Instead of incorporating middlewares into lower layers that enable vertical market integration, another architectural alternative is to add a business layer on top. The top market layer uses a service-oriented approach to provide application availability, sharing, and cross-vendor deployment as necessary [59,60]. These models are mostly service oriented in design, and they extend object extractions to middleware service management layers. Thus, the application and business layers enable intelligence and integration of the SOA-based fabric into vertical markets. Al-Fuqaha et al. [48] investigated various architecture models and presented a comparison of a few architectures based on multiple layers toward the need to design a reference architecture that provides scalability, interoperability, and easier integration.

R. Khan et al. [53] proposed a basic IoT architecture model where the three-layered approach aligns the IoT fabric close to the internet stack. The three layers, namely Perception, Network and Application, as presented in Figure 6, outline the technology and device-level information in the perception layer. The network layer is responsible for the transport or communication of this information to the upper layers. The application layer is a unique blend of managing the data and scaling vertical application-specific integration in multiple domains.

Modern internet-based systems are made up of billions of devices with varying components, modules, operating systems, and thus standards implementation. Due to limited equipment resources and energy usage requirements, the introduction of lightweight protocols and specifications is a common trend in IoT-based systems. This deviation from the standard IP-based implementation necessitates a thorough examination of various technology layers. This section provided a reference IoT architecture model with device, network, and cloud layers, with components and processes decoupled at each layer. There are also several other reference models that support a layered architecture. Support for middlewares and enterprise business layers to enable vertical market integration is one of the most notable developments in comparison to IoT architectures. Reference IoT architectures were found to be mostly aligned with application-specific markets. In addition, the reference models proposed a layered architecture that can be mapped to the OSI layer models. However, a tradeoff in terms of complexity and scalability is on the horizon, limiting the standardisation of IoT architecture even further.

## 5. Understanding IoT Functional Blocks

The IoT reference models are a step forward in understanding the availability of resources, technologies, and business layer convergence in order to ensure vertical market scalability. Similarly, the roles of the various elements operating on these layers must be investigated. In this article, we present a novel viewpoint that decouples the fundamental building blocks of IoT (identification, sensing and networking, computation, services, and analytics) from their operations in different architectural layers as shown in Figure 7.

Sensing, communication and computation rely on the underlying hardware platforms, whereas, the remainder of the blocks cover middleware, relay networks, and cloud computing infrastructure, most of which constitute the vertical market. These building blocks are now thoroughly investigated in the following sections.

### 5.1. Identification

Typically, data streams on the Internet are aggregated to monitor overall network traffic. To successfully provision network resources and security policies on an IP-based network, source identification is required. Traditional internet traffic relied on the IPv4 addressing scheme, which is gradually giving way to the IPV6 addressing scheme. The IP stack implementation in IoT devices differs from that of traditional Internet nodes. The Internet endpoint typically implements the entire TCP/IP stack, whereas IoT applications frequently use a lightweight protocol implementation. With the number of IoT devices expected to exceed 75 billion by 2025, the IPv4 addressing scheme may be insufficient. As a result, almost every IoT implementation now uses the IPv6 addressing scheme. IEEE protocols, such as 6LowPAN (IPv6 Low-power wireless Personal Area Network), provide a full-stack IPv6 protocol implementation alternative in low-power devices [61,62]. It enables encapsulation and header compression, reducing network overhead while providing unique node addresses for billions of devices.

Another important factor to consider when addressing IoT nodes is the various roles that the node may take on (such as sensing, actuation, relaying or edge-gateway). The aggregation of information based on common service roles is very popular, so nodes must be identified using their Service IDs. An IPv6 address can only identify a node in the network and cannot provide additional information about its roles or behaviours. Therefore, service tag identification is also required for IoT nodes [63]. Radio Frequency Identification (RFID) provides a typical example of service identification by assigning Electronic Product Codes (EPC) to differentiate between various objects and services [64,65].

IP-based networks are becoming very large and complex as IoT applications grow at such a rapid pace. A high volume of network traffic on a single route can degrade overall network traffic, causing congestion and bottlenecks. It is therefore critical to logically slice a larger network in order to ensure faster and more reliable network traffic routes. Traditional IP-based networks logically subnetted a large network by using a private addressing scheme and techniques like Network Address Translation (NAT). Maintaining private routing information for billions of IoT devices, on the other hand, is a challenge. IoT edge-gateway devices may be unable to maintain routing and translation tables due to limited on-board computing and networking resources.

A lot of researchers are investigating smart addressing schemes for these devices. One such interesting addressing mechanism is proposed by Moghadam et al. [4], where traditional EPC codes are mapped over the IPv6 addressing scheme to uniquely identify IoT objects. The mapping technique discussed in this research focuses on creating a unique IPv6 address for every unique EPC code assigned to an IoT node based on its functionality. However, one may argue that the proposed scheme may not scale well, as multiple addresses are required to identify the same node. Hirotsugu Seiki et al. [66] presented a unique concept of a de-centralizing blockchain-based μCode management system that can also be implemented on IoT nodes, allowing scalability and a unique addressing scheme.

### 5.2. Sensing

IoT networks sense, aggregate, and relay data from billions of smart objects all over the world. In cloud data warehouses, a large amount of raw data is stored and analysed. The knowledge gained from the data assists in the introduction of business intelligence in order to make informed decisions. IoT sensors can be deployed as individual devices (actuators, smart sensors, smart wearables) or as a network of devices (such as WSN) that perform a similar function collectively [67].

WSNs are commonly used in military and industrial research applications where a large number of IoT nodes sense, collect, and relay data collectively. The roles/behaviors of on-board sensors may differ depending on the application. It is also common to see IoT nodes equipped with a variety of sensors that can be triggered based on the situation. These smart sensors’ operational requirements (such as degree of accuracy and/or power consumption) may also vary. Figure 8 provides an overview of a range of sensors typically deployed in industrial applications.

Most of the sensors are passive in nature and require a hardware platform to process the input. Usually a Single Board Computer (SBCs) is capable of processing the information on-board. These SBCs then use the communication tools and built-in TCP/IP stack to connect these nodes to the internet. Currently, we have several IoT-enabled plug and play platforms available that are extensively used in IoT research (e.g., Arduino, Raspberry Pi, Galileo, and BeagleBone). These devices are typically deployed as a single sensing node or in a mesh network topology as the sensing requirements grow. These nodes can be programmed to relay data within their network or to connect with a central management portal where the data is offloaded, analyzed, and then presented on custom dashboards.

IoT nodes are energy-constrained devices that require optimal use of on-board computing to conserve energy. Another consideration that regulates the use of energy-efficient sensors is the overall price of the IoT device. A typical node usually integrates passive sensing devices to reduce its cost. Therefore, smart sensing schemes [69] are required to conserve and, in some cases, harvest energy for IoT nodes [70,71]. In addition, IoT devices generate exabytes of data every day, which is relayed to data warehouses for processing. Researchers have been investigating ubiquitous energy autonomy and compressing schemes to provide smart sensing in recent years [71,72].

Energy autonomous compressing schemes aggregate the sensor data and relay the critical information to the data warehouses. Amarlingam et al. [73] presented an exciting compressed sensing technique for WSNs. The fundamental observation was based on treating the WSN network as the sensing node for IoT networks. The researchers presented the concept of dictionary learning of data over the wireless nodes that could be aggregated to save energy by choosing the minimal transmission cost path from the data to the sink. Such compressed sensing techniques can significantly help to save energy resources over large-scale deployments.

### 5.3. Communication

IoT networks are a combination of heterogeneous smart objects, communication technologies, and protocols that collectively perform application-specific tasks. Most of the IoT networks are built on top of the WSN that use low power wireless communications [74]. The IoT nodes must run in low power modes during their lifespan due to energy restrictions. Low power radios and the (noisy) wireless channel contribute to their architecture and working complexities. A typical communication protocol must provide instructions on data coding, transmission and flow controls, sequencing, and error correction. The hardware stack implements these protocols to develop basic applications and interfaces to transmit the data towards the target device. There are several technologies for IoT communication, which currently range from close-to-short range communications. Examples of communications technologies commonly used in IoT networks include Near Field Communication (NFC), Narrowband IoT (NB-IoT), Ultra-Wide Bandwidth (UWB), LTE-A, WiMax, WiFi and LoRaWAN [75]. Table 2 compares some of the most utilised communication technologies in IoT networks.

RFID Technology has been extensively used in the last decade for M2M autonomous communication. Several RFID-based systems (active, passive, and hybrid) for object identification and communication have been developed in the past. RFID systems operate on a data query signal that returns locally stored object information from the reader. RFID-based systems are mostly passive, offering a low-transmission rate over a short range [77]. Logistics and warehousing is one of these industry applications [78]. Some very fascinating IoT implementations focused on RFID have now emerged, including pervasive RFID installations for real-time information systems [79].

Intelligent healthcare systems are one of the most active research subjects in electronics, bioengineering, and computer science [80,81]. The current expenditure on Intelligent Health Care is about $7 billion annually. Amendola et al. [82] presented an environment of a health care system that included the implementation of body-centric wearable RFID tags to track the patient’s motion and environmental features autonomously, generating a real-time knowledge database. Wearable smart devices (smart tags, fitness trackers, smartwatches) are increasingly being deployed for health and personal activity monitoring. A wearable fitness tracker is mainly concerned with sensing personal activity and computing a fitness rating. Low cost and low power consumption are some of the common design considerations of wearable devices. Many of these devices combine identical sensors and computational algorithms; there are considerable wireless technologies available. In [83], Fan Wu et al. presented an interesting comparison in wireless technology that is suitable for wearable nodes. Additionally, some of the widely deployed wireless radio technologies for IoT applications are presented in Table 3.

LoRa is another networking technology that operates over longer transmission distances and consumes low power. The lower cost compared to cellular and WiFi systems is a significant efficiency factor for LoRa. LoRa-based IoT systems are commonly used for long-range communication where high transmission speeds are not required. One such design and evaluation environment is presented by H. Lee [84], where LoRa is chosen as the communication technology for a mesh network of IoT sensor nodes. The research also proposed an architecture for improved coverage, with fewer gateways to reduce complexity and overall deployment costs.

Cellular communication, especially 3G, 4G, LTE, and LTE-A (including the prospects of 5G technologies), is thought of as a significant communication technology for IoT applications that demand multimedia transfer or streaming capabilities [85]. Current research on 5G systems supports a design trend that enables LTE-A and beyond cellular technologies as major backbone network contenders for extended coverage, high throughput, and lower latencies [85,86]. Although, by its design, the IoT architecture is heterogeneous, it seems nearly impossible to use a single communication technology implementation for IoT applications.

### 5.4. Compute

Computation is the process of calculating the system workload (both arithmetic and non-arithmetic) for a set of I/O instructions. In traditional computer technology, microprocessors handle the computing load in their CPU cores. High computation and efficiency tasks necessitated the implementation of GPU in order to form computational clusters. IoT systems are primarily governed by a number of factors, including device costs, lower calculation loads, and low energy usage, necessitating the use of power and cost-effective processing units (microprocessors, microcontrollers, System on Chip (SOC), and Field Programmable Gate-Arrays (FPGAs)). The majority of IoT deployments include sensing, processing, and relaying data to the cloud for intensive calculations. These requirements necessitate a limited on-board processor and storage space, thereby reducing system costs and complexity. CPU optimization is frequently essential when the idle CPU wastes a significant amount of computing power. An under-provisioned CPU, on the other hand, will quickly reach its processing peak, necessitating more resources and potentially causing processing bottlenecks. As a result, workload optimization is critical and is taken into account during the initial design stages.

Traditionally, workload optimization is performed by the CPU and is typically managed by the device’s operating system. However, as the number of IoT devices grows year after year, it is clear that a heterogeneous approach is required, with edge computing playing a significant role. In comparison to low-processing, node-level IoT devices, such a solution necessitates heavy computing devices. Processing platforms such as (Arduino, Raspberry Pi, Intel IoT Development Boards, BeagleBone, and ARM Corstone) provide a range of processing capabilities from low-power to full application-specific SoC platforms. In terms of programming, the hardware is only as good as the operating system (OS) that it runs. IoT OS enables devices and applications to communicate with one another as well as other devices such as cloud networks and utilities. The IoT OS also handles the computing power and other services required to collect, transfer, and store data, essentially acting as the device’s central nervous system.

We now have a plethora of IoT operating systems that can perform a variety of tasks on a variety of hardware platforms. IoT OS examples include Contiki, RIOT, FreeRTOS, Mbed, TinyOS, Windows 10 IoT, and Zephyr [87,88]. A few of the major considerations when selecting an IoT OS are based on architecture, programming model, process scheduling, and hardware platform support. Table 4 lists some of these design specifications, whereas, IoT hardware platforms are discussed in Table 5. Leading IoT solution providers (such as IBM BlueMix and AzureIoT) also support device emulation and virtualization, which aids in the deployment of application-specific solutions [87]. A simulation environment enables researchers to build and simulate real-world scenarios using various topologies, which saves time and money. There are several cloud platforms that extend their network and application connectivity via the Platform as a Service (PaaS) model to host IoT software.

The cloud infrastructure model for IoT devices has matured in recent years. Market leaders such as Amazon IoT, Google Cloud Server, ThingWorx, and IBM Watson are attempting to turn their IoT platform (PaaS) into a service. Cloud systems, as opposed to traditional IoT networks, have greater computational, storage, redundancy, and analysis capabilities. Many exciting solutions (augmented by the cloud computing model) for real-world challenges, such as traffic management in the Internet of Vehicles, are proposed (IoV) [89,90]. It is one of the fastest growing areas of research that relies entirely on high-speed cloud computing. The Open Automotive Alliance (OAA) is investing a great deal of technical resources to realise an IoV-based Intelligent Transportation System (ITS) [91,92]. It is worth mentioning that future IoT systems need hybrid computing capabilities ranging from low-power IoT nodes to mid-end gateways to high-compute cloud networks.

### 5.5. Services

The Internet of Things (IoT) is a vast network of heterogenous devices, infrastructure, protocols, and applications. In the previous section, we looked at several IoT architectures that provide a variety of services for vertical market integration. SOAs (Service Oriented Architectures) provide an abstraction layer that connects objects to application layers. The underlying device complexities in the technology stack are greatly simplified by an abstraction layer, allowing for easier vertical market integration. However, the addition of an abstraction layer alters the system architecture at both the OS and Middleware levels. A few researchers also proposed simplifying the architecture by integrating web services directly into IoT sensors [93,94]. This strategy not only eliminates the need for abstraction, but it also ensures rapid production and deployment.

Although several service models are available, the long-term objective is to incorporate scalability, interoperability, and easier market integration. However, the categorization will distinguish between different IoT service industries and service models in terms of their services. Mathew Gigli et al. [95] presented an IoT service model based on four categories, including, identity-related services, information aggregation services, collaborative-aware services, and ubiquitous services. These categories use processes that integrate components into various layers of the technology stack. The object identification service helps to sense and identify the virtual object, which is passed on to the information aggregation layer for data aggregation. Collaborative-awareness is achieved by aggregation of information gathered from similar service profile end devices. The omnipresence of IoT intelligence is the desired end goal that can be accomplished by creating collaborative-conscious services that are intelligent enough to make automated decisions. Applications that require user-related data (such as in banking, health care, smart homes) may contain confidential information that can enable user-tailored ubiquitous services. However, the actual implementation of ubiquitous services still seems complicated and challenging.

A category-specific application analysis is used to highlight the scope of these Services in the following context. The goal is to link the literature to application scenarios in which different services stack up to achieve ubiquitous computing. Intelligent automation and orchestration are expected to be features of future ubiquitous IoT environments. RFID-based technologies are frequently used by early adopters of WSN and IoT-based technologies, including application domains such as logistics, digital storage, and fleet management. M2M systems, which include mobile, GPS, and internet technology, have been used in recent years to automate logistics operations and track goods in real time [96,97]. These examples classify applications on the basis of identity-related classes in the service model. Some of the top developments in the logistics industry, where identity-related information is mainly used, are location management systems and inventory and tracking systems [98,99]. Intel reports that almost 30% of perishable produce from farms never make it to the markets [100]. Object and inventory tracking can help track products such as farm produce, pharmaceuticals, and industrial chemicals in real-time, thus saving a fortune annually.

Logistics 4.0 enables to monitor the shipment quality and object tracking in real-time. Real-time logistics tracking systems are commonly used in the transportation of pharmaceuticals, industrial hazardous chemicals, and life-saving drugs [101,102]. On-board sensors transmit information on shipping in real-time, which lets companies not only monitor items but also ensure the optimal consistency of handling procedures. Next, ITS systems are known to be a core component of the information aggregation class. These networks include a range of subsystems, such as smart parking, smart roads, traffic management, and control systems. The sensor information from these subsystems is aggregated to ensure a consistent and safe transport experience. The aggregated information is then passed to the upper collaborative-aware services model in order to thoroughly investigate the data and make smart decisions [97]. Connected vehicles utilize this concept to leverage a collaborative-aware service model, to make real-time collaborative decisions. Driver-less cars use this service model to adapt to road and weather conditions, avoid highway congestion, and book parking spaces in advance [89,103].

Connected vehicles or smart vehicles use this concept to leverage information gathered from all of the modules mentioned above and present it to the cloud via a Collaborative-aware service mode, where real-time decisions are made, allowing for the driverless self-driving vision of cars to become a reality. Google is widely regarded as a pioneer in the development and deployment of self-driving vehicles. It creates a collaborative-aware experience for driverless cars by combining user-generated data and cloud-based machine learning models [104]. The technology has advanced so much that autonomous cars have reportedly already driven more than 4 million miles [105]. Scientific standardisation is constantly updating its blueprint to keep up with such fast-paced research and deployments. IEEE 802.11p included amendments to support short-range communication vehicles.

Wireless Access for Vehicular Environments (WAVE) [106] is an approved standard for ITS systems. The National Highway Traffic Safety Administration, USA (NHTSA) has been working closely with research organizations, standards bodies, and academic institutions to advance the goal of Vehicle-to-Vehicle (V2V) communications [107]. Researchers investigated the self-configuration capabilities of future IoV systems that support the standardized technologies and protocols [108]. In a recent report on the readiness of V2V Technology, NHSTA reported a decline in the annual deployment costs of V2V technology. The study further proposed that cross-industry standardization not only decreases the cost of development but also assures a quick rollout. Standardization also helps eliminate loopholes in the area of transportation which can be crucial, and in some cases, life-saving. The report also stated that if V2V safety applications are adopted, it could prevent 25,000 to 592,000 car crashes annually [109]. This report provides vital statistics regarding technological advancement and the standardization efforts towards the internet of connected vehicles.

Another important IoT area that is rapidly expanding is the Industrial Internet of Things (IIoT). In most cases, IIoT systems are associated with high-volume, high-speed data streams. A typical IIoT system is ideally a low-power, small-form-factor sensor or actuator node with internet connectivity that can relay sensed data to the cloud. The cloud-based ML models are then required to automate the industrial systems in near real-time.

Xu et al. [110] proposed a trustful resource allocation and management scheme that can be wirelessly implemented on gateways as well as end devices. In contrast to traditional resource allocation schemes, the researchers proposed a hierarchical structure capable of real-time application that also ensures process privacy. Kumar et al. [111] proposed a lightweight encryption scheme that enables fast hash-keys based encryption for IoT modules in the perception layer. The proposed scheme can mitigate security risks by only allowing communication between authenticated IoT devices. While we see a substantial trend in the adoption of IIoT in Industry 4.0, the vast majority are still hesitant to integrate it due to the added layer of system dependency, increased power usage, and security concerns. In recent years, Stuxnext was the biggest cyber-security threat that IIoT systems have experienced [112]. The industrial vulnerability is a huge threat vector for cyber-security attacks, that may completely render the services non-operational and cause millions in damages [113]. Therefore, many researchers [114,115,116] are studying pro-active threat mitigation schemes for IIoT applications.

Many industrial IIoT architectures and their integration with existing automation platforms require a layer of abstraction. Siemens, one of the world’s most prominent industrial automation leaders, proposed the addition of a connectivity layer to the present automation products and technologies. This concept is integrated into its cloud-based industrial automation solution, the MindSphere [117]. Lastly, smart cities can be observed as an application of the ubiquitous services class [118,119,120]. It is a system of numerous smaller subsystems, including smart homes, smart grids, ITS and environmental response systems, that form a completely pervasive system focused on collaborative awareness [121,122,123,124].

### 5.6. Semantics and Analytics

Intelligence and autonomy at the device level are needed in order to achieve ubiquitous computing where devices can self-configure and adapt to their environment [125,126]. The autonomy of future IoT networks is projected to minimize working loads by accurately, efficiently, and smartly collecting, processing, and modelling the information [127]. A semantics framework is needed that provides granularity to distinguish between a multitude of objects and their attributes in the IoT networks [127,128,129]. Such a system helps to define and understand the correct object, and can demand the appropriate resource for the desired feature or behavior, thus acting as the central intelligence or brain of the overall operation [130].

Resource Description Framework (RDF) and its variants have been widely deployed to map attributes to the data. The semantic frameworks are used at various levels in the architecture to make the overall data trustworthy. A semantic framework for translating between different technologies and protocols may be used at the lower layers as a gateway, whereas at the higher layers it can be used for data collection. Word Wide Web Consortium (W3C) Semantic Sensor Network (SSN) ontology and annotation framework is one such example [131]. Effective XML Interchange (EXI), a lightweight representation of Extensible Markup Language (XML), is often commonly utilized for constrained devices [132].

The researchers investigated RDF frameworks to efficiently store and retrieve data from IoT devices. Rahman et al. [133] proposed a lightweight, dynamic ontology-based IoT scheme. The proposed scheme develops dynamic feature-based clusters using ML models. This abstraction of the ML-based SSN ontology scheme reduces query response latency as well as memory footprints. Padiya et al. [134] used the RDF model to analyze vast amounts of IoT-based sensor data management. They used various RDF storage mechanisms to store and retrieve data efficiently. The study also compares vertical portioning and hybrid data-aware methodology, concluding that the latter technique yields a 12% increase in results. Although the reviewers have analyzed a comprehensive data collection using various data storage and retrieval models, it is still unclear how it compares with the other EXI or JSON-LD techniques. It is also uncertain if the solution can be adequate to ensure the interoperability between various layers and systems. In general, the research model and the test bed will serve as excellent starting points for a detailed analysis of data management in IoT systems.

Hasemann et al. [135] presented a rather fascinating use-case focused on asymmetric data transfer by IoT devices. Their approach is based on IoT networks that publish a large amount of data, but receive relatively few updates. They incorporated serialization for RDF documents and opted for streaming Header-Dictionary-Triples (HDT) serialization to encrypt sensor data, resulting in a reduced data lookup table size that enables the re-use of searching entities to further save resources. Along the same lines, lightweight serialization based on RDF documentation seems to be a promising approach, since it reduces the size of data collection and can be conveniently integrated. WiseLib is one of the most common lightweight serialization frameworks for constrained heterogeneous devices. The serialization maps various device role and behaviors to the data by encoding at the device level. On the one hand, reduced table sizes can help to store and retrieve data efficiently. On the other hand, the devices use additional computational power for the encoding of information. This also opens up debates about a major study gap in the benchmarks for IoT-related energy saving schemes.

Maarala et al. [136] presented an evaluation model to test various sizes of IoT networks and corresponding semantics reasoning data. Maintaining performance, scalability, and interoperability as their primary goals, researchers calculated latencies imposed by various semantic models. During the assessment of semantic models, the researchers suggested data aggregation strategies suitable for the heterogeneous IoT network. The experimental results claim that distributed reasoning with Entity Notation (EN) formats outperforms other techniques. The results also summarized the possibility of having multiple reasoning nodes with a short EN format as the best case. The researchers also proposed that time-based aggregation produces a more stable output as compared to other strategies. Many researchers also focus on content caching schemes that reduce the energy consumption footprint for IoT networks [137].

The research provided a rigorous model for analysing emerging semantic technologies and determining the best supply cases. The effects of data formats on centralised structures, on the other hand, have not been reported. Another notable trend in their research is the similarity and uniformity in latencies as system resources or overall throughput increase, whereas this may not be the case in real-world heterogeneous IoT systems. Overall, this study is a great way to compare different semantic data formats that can be used in IoT networks.

Table 6 summarizes IoT functional blocks and associated elements, as detailed in this section.

## 6. Characterizing Middlewares for the IoT

The middleware concept facilitates development by providing a scalable interface for computing and communication that enables interoperability between various services and applications. Middleware platforms have evolved in recent years to be primarily data-centric, providing an interface for effectively managing objects and data. These platforms primarily focus on sensor networks, but in our concept of IoT architecture, as discussed in previous sections, there is a growing need for the management interface and data reflected in M2M communications. As a result, in order to clearly understand the proposed middleware platforms in this context, we define them in terms of the functions and services that satisfy the specifications outlined in the previous section. This is accompanied by a thorough examination of current middleware and data systems proposed to ensure the heterogeneity required to build scalable cloud-enabled IoT networks. Table 7 presents some of these characteristics [138].

Chaqfeh et al. [56] presented a complete overview of various middleware platforms by categorizing different application services for proposed infrastructures. Their research has broadly categorized the IoT domains, ranging from semantics infrastructure to sensor networks, to fully autonomous and pervasive robotics. Figure 9 illustrates the data flow within various applications with or without the integration of middleware.

Despite the fact that the specifics of IoT functional blocks were fine-grained in the previous section, the following specifications are provided based on the implementation requirements and the IoT infrastructure software layer [139]. In this study, we look at middleware technologies based on the categories of security, privacy, and trust, which are the most important open research areas in this field. Similarly, Razzaque et al. [138] discussed that to lay out the fundamental requirements for middleware platforms, it is critical to understand the characteristics of IoT infrastructure and the characteristics of IoT applications. Table 8 presents some of the evaluated approaches that addressed these challenges.

Security and privacy are the most important areas of research that have yet to be addressed. The middleware interfaces the technology stack with the vertical market. Hence, it is of utmost importance to resolve data protection and privacy issues in this layer [8,140]. Various middleware characteristics and their impacts on user data are discussed in this study. It is very important to categorize these elements on the basis of specifications (such as infrastructure and application requirements) that will ensure potential standardization and interoperability [141]. Along the same lines, Ngu et al. [142] presented various IoT middleware use-cases by analyzing various architectures in the context of security, privacy, and trust.

Fortino et al. [143] proposed a Smart Objects (SO) based architecture that incorporates agent-based computing to support distributed deployments, whereas a backing cloud architecture performs the heavy lifting with flexible and scalable cloud compute resources. The researchers focused on existing middleware and cloud technologies to implement this multi-tier extension to examine future deployment possibilities. There is currently a broad variety of IoT middleware and development frameworks that differ in their implementation and architectural layout. However, in comparison to open source consortia, the major share is based on enterprise middlewares that are modular to service-based technology solutions [142].

C. Perera et al. [144] presented an effective and feature-rich IoT middleware that can be conveniently configured by non-IT experts. This semantic-driven architecture features a context-aware sensor configuration model (CasCoM) implemented on the Global Sensor Network (GSN) middleware. The semantics provide scalability and interoperability while reducing architectural complexities at the device level. D. Conzon et al. [145] presented a secure IoT architecture that is based on the XMPP protocol. The proposed middleware addresses both the networking and security issues by providing a secure communication channel for distributed applications. The proposed architecture allows data flow within private networks that is made secure by authentication and encryption.

In this section, we identified middleware platforms that fulfill the necessary security and privacy requirements, as well as being scalable and interoperable [146,147]. It is also worth remembering that no standard model will emerge, and IoT networks will remain heterogeneous from the physical layer to the application layer. Table 9 presents a variety of middleware architectures and technologies that address the aforementioned key issues. However, in order to protect these large-scale distributed deployments from privacy and data leaks, a standardised security engine is required.

## 7. IoT Stack Optimization

In the preceding sections, we looked at IoT architectural criteria, fundamental building blocks, and the role of middleware. The horizontal technology fabric is made up of these components, which must be optimised for interoperability, scalability, and vertical market integration. Tuning application-specific services, where hardware, software, and interfaces are configured to make the most of a specific implementation, is commonly thought of as optimization. In order to build an adaptive and scalable IoT stack that can support end-user applications, these layers must be optimised for efficiency. Because the IoT technology stack is heterogeneous, optimization entails improvements in all major layers (from PHY to APP).

At each layer, optimization may necessitate improvements in processes, components, protocols, or even technology. After a successful production and deployment cycle, traditional optimization processes are implemented. IoT domains, on the other hand, necessitate a proactive optimization strategy due to their rapid development speed and application requirements. A proactive optimization strategy ensures application-specific improvements that begin at the design stage. This would imply a larger IoT network as well as a more robust business model. From a macro perspective, IoT implementation at the enterprise level goes through various stages of maturity, which determines the amount of room for business-specific optimization [148]. These phases translate solutions from basic applications to complex ecosystems and include:First: complete vendor dependability to deploy one-off application solely run and managed by the vendors in the cloud;Intermediary: on-premise solution deployment managed by end business as well as vendors. Thus, opens room for expansion and optimization;Mature: an end-to-end ecosystem either deployed on-premise, on-cloud or a hybrid solution that demands a complete optimization of the entire IoT stack.

These different phases of IoT maturity will vary from processes to businesses. Therefore, a more robust, vendor-neutral optimization strategy is required. Figure 10 outlines this strategy by separating IoT stack layers and blocks, into which optimization is needed. To optimize the physical layer, adaptive and low-power sensors and actuators are required that can adapt to application requirements and reduce their energy footprint. It is also suggested that these devices must have the capability to be programmed from the cloud to adapt quickly to application-specific requirements. With improvements in the silicon industry and the availability of low-cost system-on-chip (SoC), it is now possible to deploy an SoC solution [149] at the nodes that is more energy-efficient and re-configurable [150].

A substantial percentage of IoT devices connect with gateway devices, which combine and transfer the data into the cloud by relaying information. With modern technology and the ability to integrate cost-effective gateway devices, the end nodes can be made more resilient and efficient by bringing the computing closer to the device. Lin and Premsankar describe how the gateway devices can optimize the technology stack and reduce the infrastructure deployment costs where a single edge gateway can service thousands of IoT nodes and maintain a reliable connection between the cloud and the devices [151,152]. IoT deployments that use wireless technologies to communicate can be made more efficient by incorporating modern low-power, long-range radios such as Lora. These low-power radios can further help to improve the life of end-nodes. Another aspect to consider is the use of Software Defined Radios (SDR) in the communication stack that can self-heal, auto-configure, and adapt the radio based on various environments [153]. With the deployment of 5G technologies, it is critical to use SDR technology, which employs critical functions such as Network Function Virtualization (NFV) to not only enable a virtualized re-configurable environment of these radios, but also to provide an opportunity to extend overall network coverage [154].

It is essential to consider middleware that enables the implementation of virtualized functions in order to utilize hardware and processes efficiently in a modular manner. In addition, many network resources can be saved by efficient network routing. Recently, researchers are investigating the use of smart routing algorithms such as Evolutionary algorithms (EA), Stochastic algorithms (SA) or, in some cases, Memetic algorithms (MA) to provide network optimization for IoT ecosystems [30,155]. One of the most severe overheads of IoT networks is the migration of data to the cloud for analysis. Heavy-lifting is still achieved in the cloud, but with Fog/Edge computing developing as a crucial option for future IoT networks, middleware will connect directly with edge networks. This would greatly minimize data aggregation and transmission delays, resulting in the bulk of processing occurring near the edge device.

Finally, the cloud-based enterprise solution requires the integration of Machine Intelligence (MI) and Deep Learning (DL) to become self-adaptive ecosystems that can improve independently. Optimizing the cloud interface, computing capabilities, and overall product life-cycle management involves the automation and orchestration of data aggregation, transmission, and analysis to create an intelligent system [156,157]. It is also essential that cloud-based solutions support virtualized modular containers so that learning and intelligence can be extended to the edge gateways. In a real sense, the optimization of the entire IoT stack is feasible if IoT devices are able to do power-efficient processing near their source. This can only be accomplished by incorporating edge computing into the horizontal IoT fabric, over which vertical markets can be incorporated.

## 8. Fog/Edge Computing: Technological Advancements, Integration Challenges and Edge-Enabled Vertical Markets

A modern IoT ecosystem envisions fast data collection, aggregation, and near real-time analytics to create intelligent and adaptive applications. The current technological advancements allow us to process massive amounts of data from billions of IoT devices in the cloud. However, as we get closer to ubiquitous computing, we will need more real-time analytics and process intelligence. The cognitive prognostics in the cloud enable machines to learn and infer from these results in order to improve end processes. At the same time, transferring exabytes of data to the cloud increases energy consumption, resource consumption, and network latency. This is where the concept of Fog/Edge computing comes into play, bringing the entire cloud processing power closer to the network edge. The Fog/Edge computing architecture shares these objectives and promises to bring the following benefits to future IoT networks, including:Reduced network latency;Enhanced compute, storage and network capacity;Increased network bandwidth;An overall increase in system response time;Privacy and node-aware security;Fault-tolerance and mitigation at node level;Energy conservation by reducing the amount of data sent to the cloud;Network robustness—by improving the network hierarchy.

These benefits are significant for future near real-time IoT systems, associated processes, and data streams. Therefore, it is essential to understand and distinguish between these concepts from an architectural and operational viewpoint. Fog and edge computing have always appeared to be inter-related in literature where a strong and definite boundary is never established. However, if we look at these technologies from a data-offloading viewpoint, we will obtain a better understanding of how these interchangeable technologies operate. Both technologies are envisioned to be deployed and used for future IoT networks interchangeably and are sometimes also referred to as mobile computing in this context [158].

It is essential to understand how these technologies relate and their differences if deployed in an IoT network, whether directly part of the network architecture or as a stub or parallel data offloading network. Fog Computing—is a concept which envisions pushing intelligence down to the local network at the gateway level. Edge Computing—on the other hand, brings cloud intelligence, computing power and storage capabilities to the local gateway as well as the device level.

It is, without doubt, that any technical advancement will come with some trade-offs or, in some scenarios, “no one-size-fits-all”. It is therefore important to examine the degree to which these technical concepts can be deployed without introducing further complexity to the system. Fog computing is defined as a horizontal system-level architecture in the IEEE 1934 specification established by the OpenFog Consortium. IEEE adopts fog computing principles for computing, storing, managing, and networking in order to enable a things-to-cloud continuum in the technology stack [159]. A reference fog/edge architectural model is presented later in this article, supporting the idea of technology integration.

### 8.1. Fog/Edge Architecture Model

A future reference architecture for IoT networks will adopt some form of N-tiered Fog/Edge deployment by loosely coupling the best of these concepts without adding additional layers of complexity to the overall system architecture. A similar approach is presented in Figure 11 by leveraging these concepts into the IoT architecture as presented in Table 10.

Several fog-based IoT architectures have been proposed to enable device-to-device (D2D) and machine-to-machine (M2M) communications. The architectures and reference models mostly include a fog and edge layer on top of the existing IoT network hierarchy. However, focusing on the challenges that the underlying technologies face from the PHY to the upper layers reveals a pattern of virtual and flexible network topologies that provide over-the-top content and management services. With a multi-tiered deployment, future IoT networks can benefit from the Fog/Edge haze. However, it is critical to strike a balance between capabilities and complexities in order to prevent the disadvantages of these concepts from overshadowing the heterogeneity and complexity of IoT networks.

Vertical markets can grow and scale on the network edges with middleware support for edge networks, paving the way for next-generation applications and services. However, in terms of network complexity, data aggregation, service value, and cost, it is arguably still the most pressing question that businesses must address: how much data must be kept on premises and on edge nodes? To address privacy and security concerns in the cloud, applications such as military, health care, and real-time response seeking applications such as intelligent transportation, ideally always require the data to be kept on-premises. These concerns can be alleviated by bringing compute, storage, and flexible networking capabilities to the network edge, but this raises another issue of edge node trustworthiness as these gateway devices become vulnerable to physical access.

Another factor that can strictly delay the introduction of Fog/Edge nodes is the additional hardware, related implementation and maintenance costs, and the complexity of the network. Table 11 extensively surveys the underlying technologies and research challenges in this domain. There is a major shift that edge enterprise technology developers have begun to concentrate on the enterprise framework rather than the infrastructure. If we allow vertical markets to expand on edge layers in terms of applications and services, there is a need for data, control, and analytics management on these layers that extends to the cloud. This can only be achieved together with the support of network operating systems and middleware to work in conjunction. However, data integrity, privacy, and trustworthiness remain open challenge in edge computing domains. With limited computing storage, and network abilities compared to the cloud infrastructure, a single point of failure, and an increased threat vector surface, the edge-enabled IoT networks appear to be more vulnerable to cyber-security attacks.

### 8.2. Security and Orchestration

To address data integrity and trustworthiness, enterprise developers present an orchestration layer within their software frameworks to deploy secure applications that can run on the edge. Unlike public clouds, fog cloudlets or edge level gateways provide public/private and hybrid cloud functionality by confining the content and data stream locally within a region that is governed by a specific security policy. The Edge networks, together with the management and orchestration interface, are required to have a distributive mechanism to counter security, privacy, and trustworthiness issues in IoT edge networks.

An interesting aspect that aims to resolve certain trust-management problems is the self-adaptiveness and collaborative computing capability of an edge computing environment. Networks can be logically partitioned across an edge gateway where the devices are geographically or spatially connected. In these environments, the communications are likely to be between object-to-object (O2O) and peer-to-peer (P2P) settings, which require either on-board or aggregated intelligence at the edge gateway level. Table 12 presents the current research on the underlying technology in the edge computing domain to address some of these concerns. Modern cellular and communication networks that support edge computing can logically partition a network segment intelligently. These processes or objects can be confined at the edge boundary level using network-function virtualization (NFV), which isolates the data stream from the rest of the network.

In a typical cloud-based IoT network, the resources, either physical or virtual, scale from “things” to the cloud. Thus, the scalability and centralized nature of these resources provide an elastic pool of resources that can be extended to devices. On the one hand, Edge computing, with its local processes, and object isolation, addresses most of the privacy and security problems cloud-computing has long endured. Isolating resources in different local networks, on the other hand, means that these compute and storage resources cannot communicate and extend to other edge boundaries, limiting their usage despite increased infrastructure. The spatial coherence, device accessibility, and environmental factors further reduce the direct access to these resources, which is also true for their up-gradation and maintenance.

These constraints pose an inherent challenge for technology developers and enterprise solutions where resource provisioning becomes the major decisive factor during deployment stages. Security policies and resource management become a further daunting challenge for edge network administrators and developers to monitor and utilize idle resources. Therefore, a unified orchestration mechanism that extends from the edge to the cloud is needed to manage the services. Cloud-based orchestration mainly emerged from automation, where a specified pool of resources is orchestrated. It is imperative to consider that the homogeneity of the resources is maintained in the cloud, which helps to scale and manage these resources uniformly. The orchestration on the network edge brings other challenges due to end-device constraints, heterogeneity of the fabric, varying connectivity technologies and, in most cases, due to device failures. Devices on the edge have different sensing capabilities, compute power, bandwidth, and deployment scenarios, making it almost impossible to know their characteristics beforehand. To envision IoT Edge network orchestration, all these parameters must be accounted for, which can be investigated during deployment staging.

Edge computing in the IoT domain faces many challenges, and the most crucial of all is the lack of standardization. The European Telecommunications Standards Institute (ETSI) presented standardization directions for edge network orchestration. ETSI Multi-access Edge Computing (MEC) standard is already making its way forward with the implementation of 5G networks by providing an ecosystem that operates at the Radio-Access Network (RAN) edge to authorize third-parties for edge vertical solutions and applications. ETSI also proposed an open-source Network Functions Management and Orchestration (MANO) software stack closely aligned with the Network Functions Virtualization (NFV) framework. MANO, together with edge operators running MEC, can utilize additional service layer orchestration to realize a unified orchestration that runs from the network edge to the cloud.

## 9. Discussion

The IoT domain has seen constant growth in its technology and adoption. The WSNs provided a fundamental infrastructure for IoT applications that later included AmI through the integration of pervasive CPS. This trend is expected to experience exponential growth to achieve a truly ubiquitous computing platform. The rapid development cycle, market requirements, and overall revenue make IoT domain a trillion dollar industry. This assures a potential goal of merging diverse IoT ecosystems to create a genuinely smart digital world. However, IoT innovations are seen as some of the most heterogeneous in existence due to the lack of standardization. On the one hand, technological advancements in the electronic industry enabled cost-effective, rapid IoT production. On the other hand, the IoT market is strongly tied into various vendor-specific ecosystems. This research article aimed to closely investigate the IoT technology stack to outline current research challenges, optimization opportunities, and frameworks to improve vertical market integration.

The first step in understanding IoT environments is to study their architecture closely. We examined a number of reference architectures that allow effective technology deployment as well as market integration. IoT systems, at their heart, are supposed to collect data, which is then analyzed to generate business insights. Therefore, most of the proposed reference architectures involve a business adaptation layer that helps to integrate the technology with business applications. IoT ecosystems can vary from time-critical to mission-critical applications, but the fundamental objective is to gather and analyze information in order to make better decisions. Therefore, the future IoT platform would provide middleware and market adoption layers for smoother enterprise integration. It is therefore very important to consider that the reference architecture model must be modular, scalable, and interoperable.

Next, we examined the fundamental building blocks that operate on these architectural layers. Billions of IoT devices need a flexible and reliable addressing scheme that can define not only the device, but also the services at the same time. As these devices perform a variety of application-specific activities, it is anticipated that a robust addressing scheme is required. We have also experienced the transition to the IPv6 address scheme for Internet-connected devices. It is reasonable to expect that future IoT devices will have some kind of IPv6 implementation with service tag identification. The use of application-specific SoC platforms, which are power-efficient and cost-effective, is a crucial component of IoT implementation. Research in this area shows that energy harvesting and low-power sensing devices will dramatically decrease the total energy footprint of IoT devices. Application-specific SoC can also incorporate the required computing and communication modules to further minimize the technical complexity. Similarly, we’ve seen a number of IoT domain semantics and analytics frameworks that can allow scalable and genuinely ubiquitous systems. However, it is crucial to understand that technical developments in individual building blocks cannot be completely incorporated into a single solution, putting a strong emphasis on standardization of technology. In addition, with application-specific SoC platforms, businesses can lock end-users into their own ecosystems, which would further jeopardise growth and interoperability.

The use of middlewares is an essential prerequisite to align the application stack with vertical market solutions. IoT middleware connects the physical (hardware layer) and virtual (application layer) worlds of devices and information management systems. It provides the necessary features to simplify inter-process communication and to provide an abstraction layer for connecting to business solutions. In other words, it is mediator interface, a software layer between the “internet” and the “things”. Among the most critical performance features for middleware platform design are interoperability, platform independence and portability, scalability, and security. Security, trust and governance are some of the most important problems that middleware is facing. It is very difficult to find a balance between use and functionality. Middlewares revolve around processes, their data and their integration, thus, must have inherent security.

Because middleware serves as an abstraction layer, having an open source framework for middleware that can scale in both directions is ideal. Future IoT networks require middle-ware architectures, interoperability with enterprise applications, open data format support, and multi-service heterogeneous device support are all required to build future IoT networks. In this article, various middleware characteristics that are based on infrastructure as well as application requirements are presented. Towards the end, we listed some of the current and market-leading middleware platforms that either do not completely meet security needs or are too complicated to be integrated into current solutions. Therefore, the current middleware goals and security aspirations are characterized for future IoT systems.

In Section 7, the IoT stack optimization opportunities and trends are discussed. It is essential to consider that significant improvements can be made in the entire IoT stack. However, the gateway devices and network layer provide a vast playground where many improvements are required, either based on the current technologies or by integrating modern solutions and technologies into the existing infrastructure. It is also essential to consider that current cloud-based solutions need optimization by incorporating AI-based ML and DL models that will not only optimize the process and end solutions in the cloud, but will also help to bring the same intelligence to the gateway devices, thus making the network and edge layer of the technology stack more powerful.

Section 8 presented a detailed insight into fast-emerging edge computing networks for IoT that promise to extend cloud resources to the network edge. The section presented a unique perspective on underlying technologies and research in edge computing and how to build vertical applications and services on the network edge. Fog/Edge computing presents a promising future for the IoT domain, with added functionality and infrastructure to achieve ubiquitous computing power. The idea of segmenting, isolating, scaling, and extending network resources between different islands bordered by processes, objects, or security policies is fascinating. However, a clear proof of concept is still missing.

Despite the fact that the Internet of Things has been making waves for over a decade, we must admit that a typical IoT node is getting old. Future IoT networks will bring intelligence and computational power from the cloud to the farthest reaches of the Internet, thanks to the availability of more capable hardware at lower prices. Because all processing was done in the cloud, traditional IoT networks with limited on-board computational power experienced network latencies. Data off-loading causes network latency in the typical integration of IoT devices with edge networks. We can now take real-world actions with sub-millisecond latencies thanks to modern edge devices with plenty of processing power. In addition, compared to traditional IoT nodes, we can collect more samples and feature rich data thanks to the extended edge device capability. In a nutshell, edge computing analyses some of the data from IoT devices at the edge of the local network rather than sending it to the cloud, resulting in faster, redundant, and scalable IoT processing. However, replacing traditional and cost-effective IoT nodes with edge devices is not financially feasible. As a result, a middle ground must be established to ensure that these heterogeneous networks operate optimally and provide near-real-time experiences.

Edge computing has many uses in today’s IoT world. Due to latency and connectivity issues, some IoT use cases are impossible without such a distributed, local computing framework. This essential technology is the backbone of IoT applications that use classified data, involve significant or low-latency decision making, and operate in environments where cloud connectivity is limited or nonexistent. Edge computing is critical in industrial IoT scenarios, such as on a factory floor, to reduce downtime and data breaches while improving data management. Furthermore, IoT integration with edge computing networks ensures increased bandwidth and data integrity, resulting in higher resolution datasets. Finally, integrating edge computing with IoT networks will improve security and privacy. Edge computing is unlikely to be more secure than a private cloud, but it is more local. In-house edge servers ensure that data never leaves the company’s local perimeter and control all access to the data storage servers. However, edge providers must meet a few IoT-specific requirements. Some IoT applications that require huge AI or business intelligence at the edge may require specific hardware, such as GPU-capable servers. Edge gateways enabling IoT devices will need to support multiple device communication protocols like ZigBee, Bluetooth, cellular, and Wi-Fi. Operators should also think about IoT ecosystem they are using and how it reinforces and integrates with their edge network.

The lack of standardization, privacy and trust-management concerns, and the need for distributive orchestration that extends from edge to cloud are still some of the biggest challenges in this domain and seriously hamper the growth of enterprise solutions in vertical edge markets. Ubiquitous computing and the future vision of achieving interconnected IoT systems are promising, yet far from being achieved. We expect device miniaturization and platform availability for future IoT applications. However, without a standard architecture, technology blueprint, and integration readiness, the overall IoT heterogeneity will only increase.

Future research could look into the open issues in IoT to see if there are any new approaches that could be used to solve them. A truly innovative middleware research design may also be proposed, incorporating a fresh outlook for managing smart objects and applications, as well as a solution for unsolved outstanding issues in a particular area of expertise, such as security, privacy, and interoperability. In addition, the integration of traditional IoT networks with edge networks, as well as the use of edge computing devices for resource enhancement, must be investigated. Future research on next-generation IoT networks should look into open-source middleware abstraction layers that support the integration of IoT and edge devices. Aspects of data management, such as data offloading on edge devices and inherent data security measures, must also be carefully considered.

## Figures and Tables

**Figure 1 sensors-22-00995-f001:**
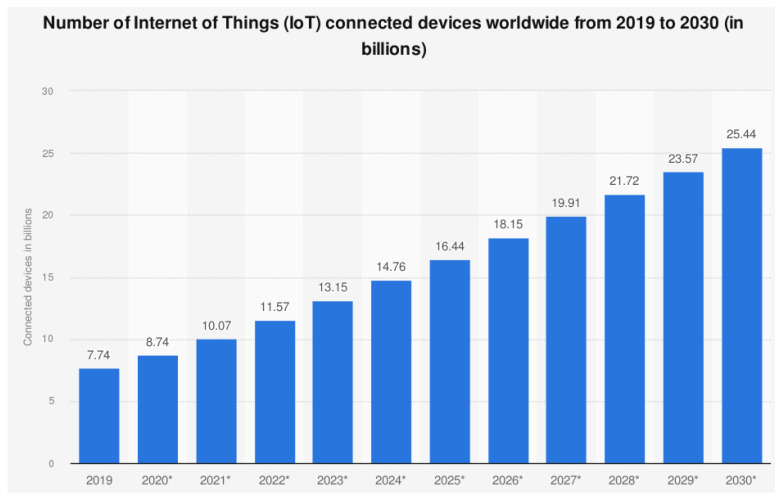
Internet of Things (IoT) devices installation growth trend [1]. Asteriks means projection year.

**Figure 2 sensors-22-00995-f002:**
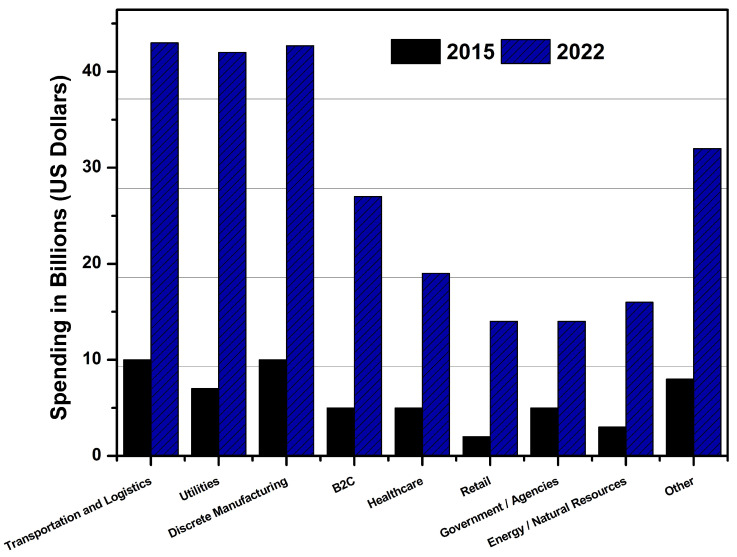
Market spending projection (USD billion) in various IoT industry sectors.

**Figure 3 sensors-22-00995-f003:**
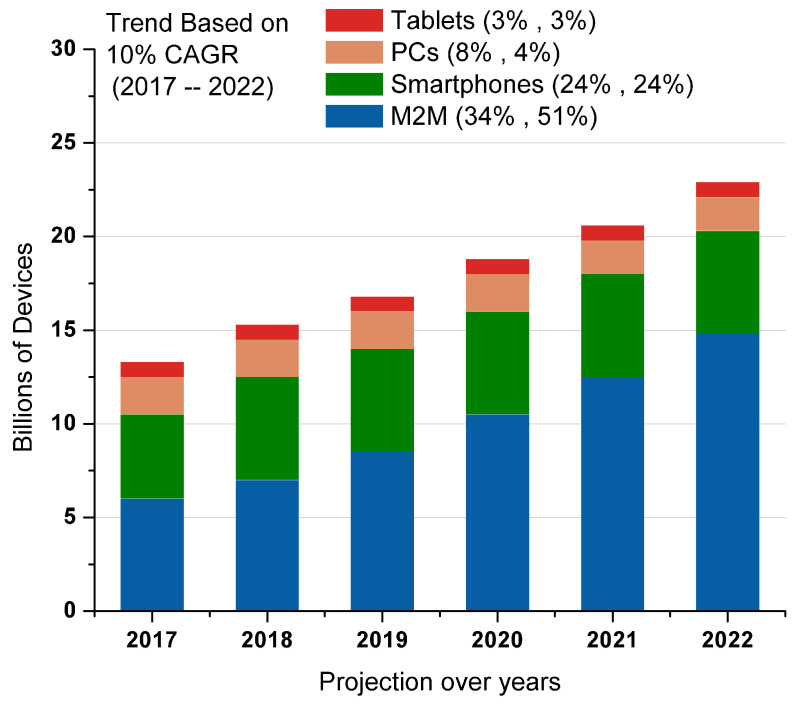
IoT traffic trends for M2M communication over next 5 years.

**Figure 4 sensors-22-00995-f004:**
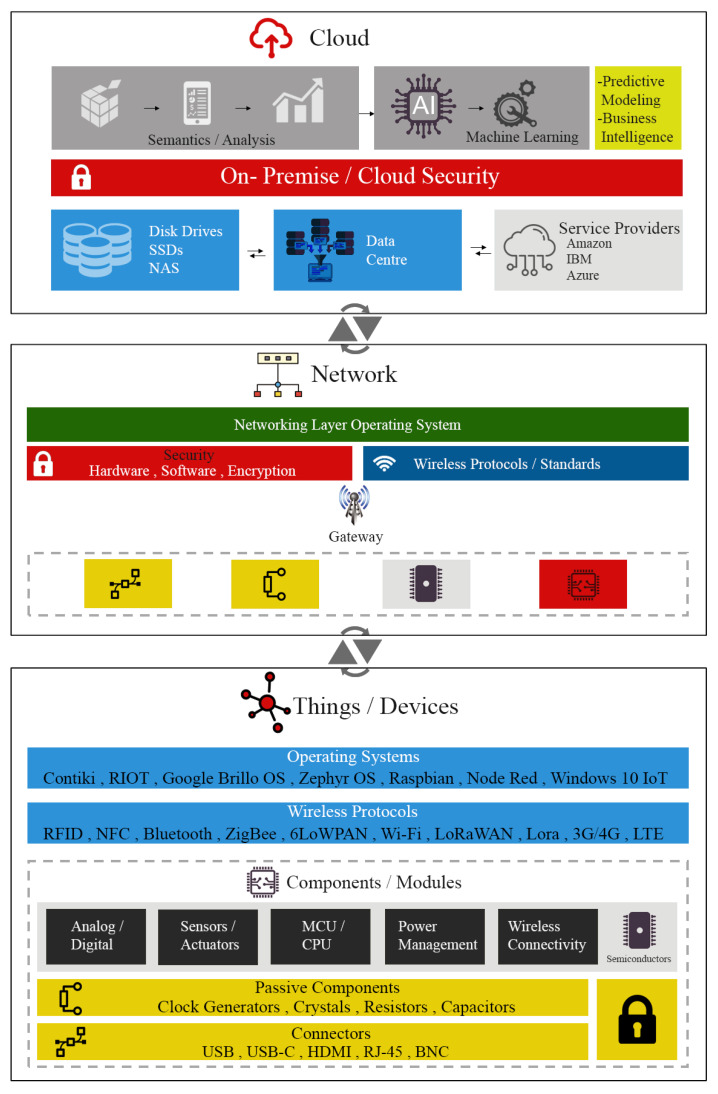
IoT architecture model: technology fabric from physical (PHY) to application (APP) layers.

**Figure 5 sensors-22-00995-f005:**
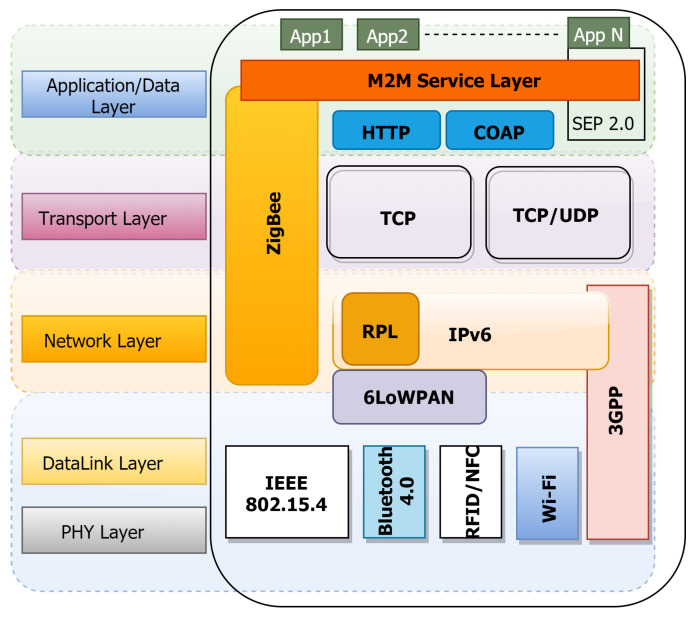
BUTLER EU Project—layered IoT architecture model.

**Figure 6 sensors-22-00995-f006:**
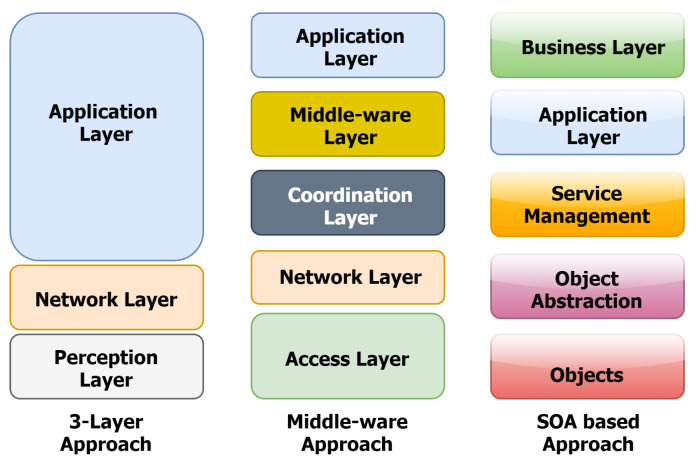
The IoT architecture: layered model approach.

**Figure 7 sensors-22-00995-f007:**
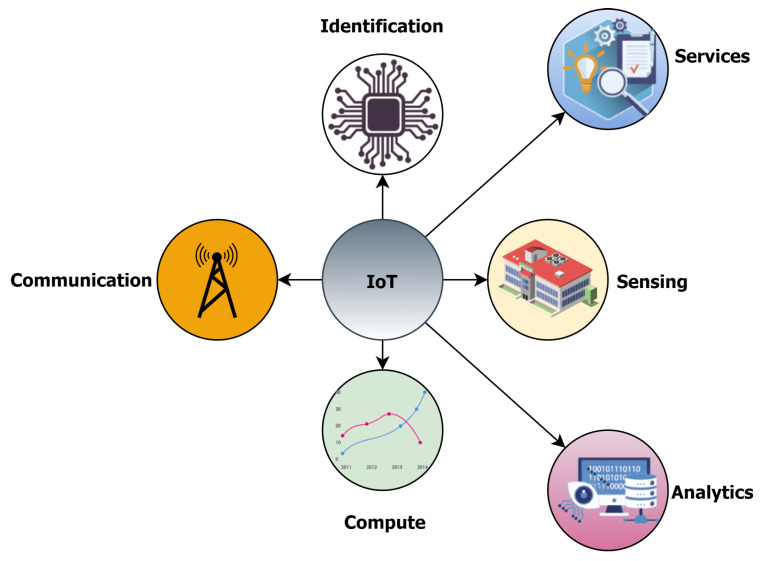
The IoT functional elements.

**Figure 8 sensors-22-00995-f008:**
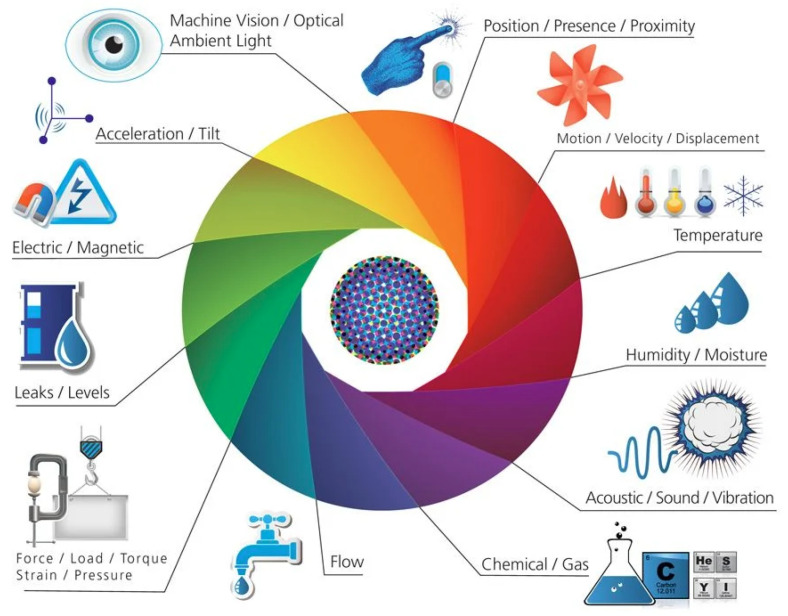
Sensors and actuators currently deployed in the IoT domain [68].

**Figure 9 sensors-22-00995-f009:**
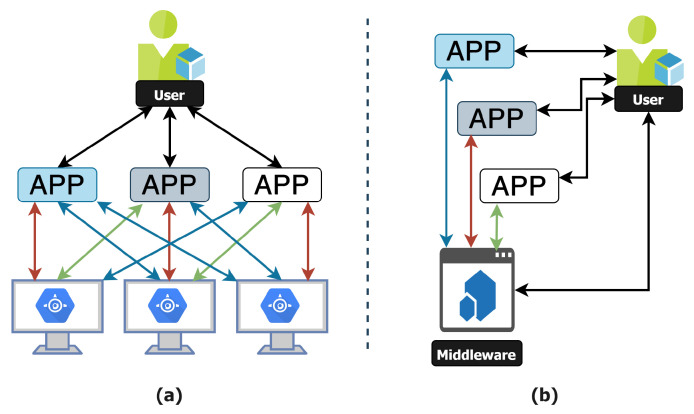
(**a**) Illustration of user data flow without middleware. (**b**) Illustration of middleware integrated IoT network responsible for handling data flow between users and multiple applications.

**Figure 10 sensors-22-00995-f010:**
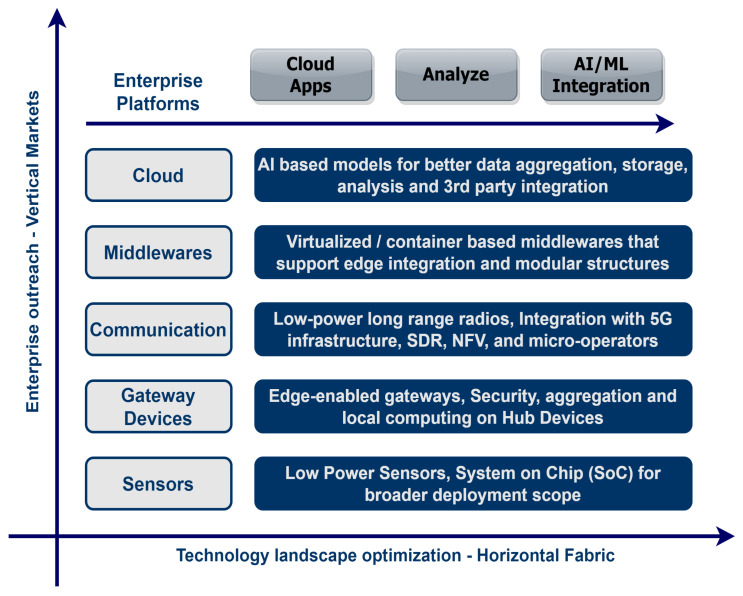
Proposed approaches for optimizing the entire IoT technology stack.

**Figure 11 sensors-22-00995-f011:**
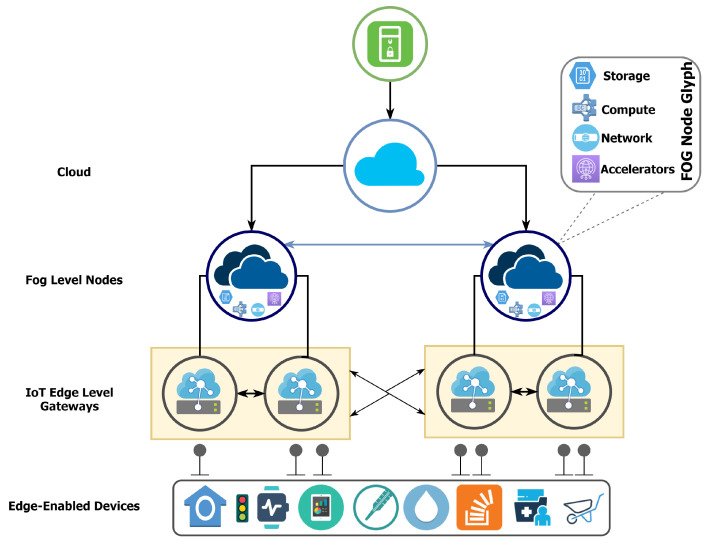
A multi-tiered Fog/Edge level architecture for the Internet of Things (IoT).

**Table 1 sensors-22-00995-t001:** Review/survey papers and their contributions in IoT application domains.

Year	Article	Title	Major Contributions
2021	[18]	Internet of Things (IoT): A Review of Its Enabling Technologies in Healthcare Applications, Standards Protocols, Security, and Market Opportunities	Investigation of security, privacy, and Quality of Services (QoS) in IoT based healthcare applications.
2021	[19]	Blockchain for IoT-Based Healthcare: Background, Consensus, Platforms, and Use Cases	Investigation of a few methodologically presented use cases to demonstrate how key features of the IoT and blockchain can be used to support healthcare services and ecosystems.
2021	[20]	A Review of Wearable Internet-of-Things Device for Healthcare	A systematic literature review on smart wearables and its usage in an IoT health-care setting.
2021	[21]	Recent advances on IoT-assisted wearable sensor systems for healthcare monitoring	Detailed investigation of various IoT technologies that are used in wearable and health-care environments.
2021	[22]	Edge and fog computing for IoT: A survey on current research activities & future directions	Investigation of Edge–IoT architecture environment issues including scheduling, SDN/NFV, virtualization, and security.
2021	[8]	Emerging IoT domains, current standings and open research challenges: a review	A comprehensive survey on fast emerging IoT ecosystems that require technical advancements and technology integration.
2021	[23]	A Systematic Survey on the Role of Cloud, Fog, and Edge Computing Combination in Smart Agriculture	A systematic literature review focusing on IoT, Cloud, and Edge computing in Smart-Agriculture domain.
2020	[24]	Internet of Things (IoT) for Next-Generation Smart Systems: A Review of Current Challenges, Future Trends and Prospects for Emerging 5G-IoT Scenarios	An in-depth examination of IoT technology from a bird’s eye perspective, including statistical/architectural trends, use cases, challenges, and future prospects, as well as a link between 5G and IoT scenarios.
2020	[25]	Edge-computing architectures for internet of things applications: A survey	Classification of Edge–IoT networks based on orchestration, security, and big data perspective.
2020	[26]	Overview of Edge Computing in the Agricultural Internet of Things: Key Technologies, Applications, Challenges	Edge computing in the agricultural Internet of Things is examined, as well as the use of Edge computing in conjunction with Artificial Intelligence, Blockchain, and Virtual/Augmented Reality technology.
2020	[27]	Internet of Things (IoT): Opportunities, issues and challenges towards a smart and sustainable future	Systematic research on IoT applications in sustainable environment, smart cities, e-health and AmI systems.
2020	[28]	IoT reliability: a review leading to 5 key research directions	An in-depth review for the quantification of data reliability and optimization in IoT.
2019	[29]	Intelligent positive computing with mobile, wearable, and IoT devices: Literature review and research directions	A conceptual framework for bridging the gap between IoT networks and next-generation computing services.
2019	[30]	Network optimizations in the Internet of Things: A review	State-of-the art literature survey to suggest network optimization in future IoT networks.
2018	[31]	A survey on the edge computing for the Internet of Things	Architecture-based investigation of Edge computing to enhance IoT performance
2017	[32]	Internet of things: architectures, protocols, and applications	A comprehensive literature review of IoT technologies, applications and implementation. In addition, the research provides a unique perspective in designing and optimizing future IoT systems.

**Table 2 sensors-22-00995-t002:** Enabling communication technologies for IoT networks [76].

Parameters	WiFi	WiMAX	LR-WPAN	Mobile	LoRa
Standard	IEEE 802.11 a/c/b/d/g/n	IEEE 802.16	IEEE 802.15.4 (ZigBee)	2G-GSM, CDMA, 3G-UMTS, CDMA 2000, 4G-LTE	LoRa WAN R1.0
Frequency Band	5–60 GHz	2–66 GHz	868/915 MHz, 2.4 GHz	865 MHz, 2.4 GHz	868/900 MHz
Data Rate	1 Mb/s–6.75 Gb/s	1 Mb/s–1 Gb/s(Fixed) 50–100 Mb/s (Mobile)	40–250 Kb/s	2G: 50–100 Kb/s 3G: 200 Kb/s 4G: 0.1–1 Gb/s	0.3–50 Kb/s
Range	20–100 m	<50 Km	10–20 m	Entire Cellular Coverage	<30 Km
Energy Consumption	High	Medium	Low	Medium	Very Low
Cost	High	High	Low	Medium	High

**Table 3 sensors-22-00995-t003:** Wireless radio technologies for IoT applications.

Product	Module Cost	Frequency	Range	Data Rate
STM32WL55JCI6	$11	150 MHz to 960 MHz	10 Km	~300 kbps
RFM95W	$50	430/868/915 MHz	~100 Km	~300 kbps
RFM95W	$8	430/868/915 MHz	~60 Km	~120 kbps
Sigfox S2-LP	$3	452 MHz–527 MHz, 904 MHz–1055 MHz	~50 Km	~500 kbps
CC2640P	$5	2.4 GHz	~300 m	~2 Mbps
DIGI XBEE-900HP	$50	900 MHz	~5 Km	~200 kbps

**Table 4 sensors-22-00995-t004:** IoT operating systems design characteristics.

	Contiki	TinyOS	RIOT	FreeRTOS	uClinux	Mbed
Architecture	Monolithic	Monolithic	Microkernel RTOS	Microkernel RTOS	Monolithic	Monolithic
Programming Model	Event-driven, protothreads	Event-driven	Multi-threading	Multi-threading	Multi-threading	Event-driven, single thread
Process Scheduler	Cooperative	Cooperative	Preemptive, tickless	Preemptive, tickless	Preemptive	Preemptive
Programming Languages	C	nesC	C,C++	C	C	C,C++
Supported Hardware Platform	AVR, MSP 430, ARM Cortex, PIC-32	AVR, MSP 430	AVR, MSP 430, ARM Cortex-M, x86	AVR, MSP 430, ARM, x86, 8052, Renesas	ARM 7, ARM Cortex-M	ARM Cortex-M
License	BSD	BSD	LGPLv2	modified GPL	GPLv2	Apache License 2.0

**Table 5 sensors-22-00995-t005:** Comparison of IoT supported latest hardware platforms.

Parameters	Arduino Uno Rev3	Intel Galileo Gen 2	Intel Edison	ESP8266	BeagleBone X15	Banana Pi BPI-P2 Zero	Raspberry Pi 4 B
Date Released	September 2010	10 July 2014	Q3 2014	August 2014	November 2015	July 2018	June 2019
Processor	ATmega 328 P	Intel Quark SoC X1000	Intel Quark SoC X1000	RISC based L106 32-bit	TI AM5728 2 × 1.5 GHz ARM Coretex-A15 2 × 700 MHz	H2 Quadcore Cortex-A7	Broadcom SoC BCM 2711
GPU	No	No	No	No	PowerVR Dual Core SGX544	Mali 400 MP2	Broadcom VideoCore VI
Clock Speed	16 MHz	400 MHz	100 MHz	80 MHz	800 MHz	800 MHz	800 MHz
System Memory	2 KB	256 MB	1 GB	32 KB	512 MB	512 MB	4 GB
Flash Memory	32 KB	8 MB	4 GB	80 KB	4 GB	8 GB	4 GB
Communications	IEEE 802.11 (b/g/n), IEEE 802.15.4 433RF, BLE 4.0, Ethernet, Serial	IEEE 802.11 (b/g/n), IEEE 802.15.4 433RF, BLE 4.0, Ethernet, Serial	IEEE 802.11 (b/g/n), IEEE 802.15.4 433RF, BLE 4.0, Ethernet, Serial	IEEE 802.11 (b/g/n), IEEE 802.15.4 433RF, BLE 4.0	IEEE 802.11 (b/g/n), IEEE 802.15.4, 433RF, BLE 4.0, Dual Gigabit Ethernet, Serial	IEEE 802.11 (b/g/n), IEEE 802.15.4 433RF, BLE 4.0, Ethernet, Serial	IEEE 802.11 (b/g/n/ac), IEEE 802.15.4 433RF, BLE 4.2, Ethernet, Serial
Development Environment	Arduino IDE	Arduino IDE	Arduino IDE, Eclipse, Intel XDK	Arduino, ESP Easy, Espruino	Arduino IDE, Eclipse, Cloud 9 IDE	NOOBS	NOOBS
I/O Connectivity	SPI, I2C, UART, GPIO	SPI, I2C, UART, GPIO	SPI, I2C, UART, I2S, GPIO	SPI, I2C, GPIO, UART	SPI, I2C, UART, I2S, GPIO, CAN Bus	SPI, I2C, UART, I2S, GPIO	SPI, DSI, UART, SDIO, CSI, GPIO
Programming Language	Wiring	Wiring, Wyliodrin	Wiring, C/C++, HTML5	C/C++, Python, Ruby	C/C++, Debian, Python, Ruby, Java, Shell	C/C++, Python, Java	C/C++, Python, Java, Scratch
Approximate Cost	$20	$70	$50	$4	$270	$30	$35

**Table 6 sensors-22-00995-t006:** IoT functional elements and associated technologies overview [48].

IoT Functional Elements		Standards/Technologies
Identification	Naming	EPC, μCode
	Addressing	IPV4, IPV6
Sensing		RFID Tags, Smart Sensors, Wearable sensors, embedded sensors, Compact and Low power sensors, actuators and relay sensors
Communication		RFID, NFC, UWB, NB-IoT, Bluetooth, BLE, IEEE 802.15.4, Z-Wave, WiFi, LTE-A, LoRa
Compute	Hardware	Arduino, Raspberry Pi, Beaglebone, Banana Pi, Intel Galileo, Intel Edison, Node MCU, Smartphones and Smart sensors
	Software	Operating Systems: (Windows 10 IoT), Raspbian, Contiki, TinyOS, LiteOS, Riot OS Cloud Solutions (NodeRed, NimBits, Azure IoT, IBM Watson, Kaa)
Services		Identity-related (Logistics) Information Aggregation (Intelligent Transportation) Collaborative-aware (Self-driving cars) Ubiquitous (Smart cities)
Semantics & Analytics		RDF, EN, JSON-LD, EXI

**Table 7 sensors-22-00995-t007:** IoT infrastructure and application characteristics.

Parameter	Nature	Impact
Characteristics of IoT Infrastructure		
Heterogeneity	Multi-vendor, multi-capability devices from low-cost to high-end, capable of performing heavy work	Making resources/environment dynamic, thus adding complexity for middleware to support interoperability
Resource Constraints	Small size, low power, small memory and computing capabilities	An additional challenge to implement the middleware software layer
Spontaneous Interaction	M2M communication, real-time event triggers	Automated, real-time, machine to machine interactions may require a system that is ubiquitous and requires no human intervention
Ultra large-scale Networks	Ultra-large number of events in multiples of billions every day	Event congestion, resource exhaustion, added data backups and event aggregation workload
Dynamic Network Conditions	Mesh, Ad-hoc, cellular networks or in some cases relay gateways for long-distance connectivity	Inadequate or disconnected network link outages may result in truncated, duplicated or lost data, which requires self-adjusting software to account for transmissions over such networks
Context-aware application	Spatial and temporal context from sensing nodes	Requires adaptive and autonomous behavior in software stack to analyze and interpret the data
Characteristics of IoT Applications		
Diversity	Applications range from event-driven to time-driven IoT domains	Added complexity for middleware to adapt to different application deployments providing multiple services, such as: transportation and logistics, that deploy the same hardware but demand different services
Real-time	Applications range from mission critical to time-critical IoT domains	Real-time application deployments such as in health-care, would demand an added layer of reliability and data integrity
Security	Global connectivity versus open attack surface	Small computing capability, device and network heterogeneity and a provision for global access adds complexity for middleware to mitigate security threats
Privacy	Personal versus critical data	IoT applications may contain data from health-care, financial, internal stocks to industrial deployments. The data privacy acts vary from region to region, thus adding another complexity for middleware to provide flexibility to comply with data protection acts

**Table 8 sensors-22-00995-t008:** Challenges in middleware approaches for IoT applications.

Domains			Semantic Web & Web Services				Sensor Networks & RFID			Robotics
	**Approach**		**Task Computing**	**Triple-Space Based**	**UBIWARE**	**SOA Approach**	**GSN**	**Fosstrack**	**TinyRest**	**Robotic-Based**
Challenges Addressed	Interoperability		🗸	🗸	🗸	🗸	🗸		🗸	🗸
	Scalability				🗸	🗸	🗸	🗸	🗸	
	Abstraction	I/O Hardware Devices		🗸	🗸	🗸			🗸	🗸
		H/S Interfaces			🗸	🗸				🗸
		Data Streams	🗸	🗸	🗸	🗸	🗸	🗸	🗸	🗸
		Physicality	🗸	🗸	🗸	🗸	🗸	🗸	🗸	🗸
		Development Process			🗸	🗸	🗸	🗸		
	Spontaneous Interaction			🗸	🗸	🗸	🗸	🗸	🗸	
	Unfixed Infrastructure		🗸	🗸	🗸	🗸	🗸	🗸		
	Multiplicity		🗸	🗸	🗸	🗸	🗸	🗸		
	Security and Privacy			🗸		🗸			🗸	

**Table 9 sensors-22-00995-t009:** Middleware services/platforms and their associated security models.

Platform	Technology	Addresses Security & Privacy?	Drawbacks
Service-based IoT Middleware			
Hydra/LinkSmart	Web Services, XML, Symmetric Keys using Certificate Authority (CA)	Partially, by encrypting user data	Signed certificates for billions of devices is practically impossible. No policy-based access model. No secure user data storage
GSN	Access Control	Partially, by encryption and electronic signatures	High complexity implementation. Complex query and semantics operation on data streams.
OpenIoT	Message Digests, Public/Private Key Cryptography, Flexible access controls	Fully	Generic security framework model, which is very difficult to implement. No implementation details provided for third-party applications.
Virtus	XMPP, Event-driven communications, isolation of instances	Partially, by encryption at Transport Layer using TLS and Authentication by SASL protocol	Huge payloads. Increased entity versus digest bundles.
Cloud-based IoT Middleware			
Webinos	Personal zones, Virtual user defined overlay networks	Partially, by de-coupling contextualized data, automatic filtering on personal data	Limited object access and identification outside overlay networks
ThingWorx	Query and Analysis based engine	Partially, by intelligent queries and innovative 3D data offloading	Enterprise mode. A limited number of devices can be attached, which further limits large-scale deployments of distributed networks.
Actor-based IoT Middleware			
Node-Red	Server-side scripting, event-driven flow-based approach	None. Open access to IP and ports	Vulnerable to security threats as it only provides a programming interface and does not implement security. Can only be used as a visual programming interface for rapid prototyping

**Table 10 sensors-22-00995-t010:** Edge market innovators and leaders.

Platform/Service	Edge Solution
FogHorn	The power of machine learning and advanced cognitive analytics on-premise edge
Xnor.ai	Scaled machine learning and deep learning models for edge networks
SWIM	Consistent advanced real-time device-level analytics throughout edge and cloud
Pixeom	Software-Defined Edge computing platform that extends cloud functionalities to on-premise
Deeplite	Artificial Intelligence (AI) based deep neural network optimizer from cloud to edge
Hailo	Deep learning microchips for IoT edge and Fog devices
Always.ai	A platform for developing deep learning-based computer vision applications for edge solutions
Xi IoT	AI-driven processing and real-time analytics at the edge
Zededa	Edge virtualization service to provide Industrial IoT analytics
Project EVE	An open-source edge virtualization engine allowing cloud-native application development for Edge and IoT

**Table 11 sensors-22-00995-t011:** Key research areas and technological advancements in Fog/Edge computing.

Scope	Articles	Contributions & Impact on Edge Networks
Fog based IoT Architectures	[160]	The design approach to tackle resource management for underlying cellular networks
	[161]	A high-level programming model supporting distributed, large scale fog applications
	[162]	Trust evaluation using service templates to incorporate cloud-edge computing
	[163]	Fog presence and its characteristics viability to support IoT services and vertical applications.
	[164]	M2M communications, challenges and solutions in the air interface
Bandwidth & Resource Management on Physical (PHY) layer	[165]	Disaster recovery management design of reliable virtual infrastructures to support network nodes during physical outages
	[166]	Bandwidth management and congestion control strategies for underlying communication links
	[167]	An Over-The-Top (OTT) virtual access network (VAN) architecture to support application-specific resource scheduling
	[168]	A centralized resource management scheme that is queue-aware to support fair scheduling and load-balancing
	[169]	Modeling of collective resource provisioning for mobile and cloud networks
Network selection, deployment & configuration	[170]	A congestion avoidance architecture for adaptive applications
	[171]	Hysteresis based selection and convergence of radio access technologies (RATs)
	[172]	Network bandwidth allocation based on applications as well as device priorities
	[173]	User traffic offloading based on cellular budget and future predictive usage.
	[174,175]	Proposed cache-replacement technique while offloading IoT data on to Edge networks for improved system latency.
	[176]	A mathematical model with multiple decision-making attributes for network selection
Network Inference	[177]	A network inference vision that employs relevance over the choice approach to utilize cloud backed machine learning powers
	[178]	An experimental study to outline and eliminate the human intervention in crowdsourcing applications improving inference
	[179]	Improving inferencing and associated network services by pairing network services with applications
	[180]	A framework to enable network inferencing from collaborative sensing and classification techniques for large scale mobile phone-based deployment
	[181]	An architecture to mask context-aware information in order to manage value Versus risk on sensor data
Content Management	[182]	Provided a framework to extend Telco content delivery network (CDN) with enhanced and extended control plane for future edge applications
	[183]	A framework to incorporate Content-Centric Networks (CCN) to empower the Over-The-Top (OTT) services in future IP networks
	[184]	Information-Centric Network (ICN) based IoT Middleware Architecture envisioning a unified IoT platform
	[185]	A distributed name resolution scheme for future Information-Centric Networks (ICN)
	[186]	An insight into software-defined network coupled with network functions virtualization for future Fog based networks
Edge Analytics & Data Mining	[187]	A mobile sensing, efficient task distribution and adaptive platform that can be utilized on Edge networks
	[188]	An adaptive cloud-based resource rate selection algorithm to support real-time stream mining applications on the edge
	[189]	An improved edge cloud framework model featuring virtualization, edge computing and local traffic offloading
	[190]	A comprehensive review of data stream mining challenges and available techniques
	[191]	A distributed dynamic data-driven mining scheme for adaptive edge vertical applications
Security, Privacy & Trustworth- iness	[192]	An insight into the reliability aspect of the network extending from cloud to edge networks
	[193]	A model framework based on offensive decoy to mitigate data attacks on the resident data in the cloud and fog networks
	[194]	Third-party auditing based public data integrity auditing scheme with no exposure to content in the clouds
	[195]	A light-weight privacy preservation data aggregation scheme for hybrid heterogeneous IoT based networks
	[196]	A distributed Block-chain based software-defined network architecture to run on Fog nodes

**Table 12 sensors-22-00995-t012:** State-of-the-art research on mitigating security concerns of Edge networks.

Scope	Articles	Major Contribution
Resource Management	[197]	Radio and Computational resource management in Mobile Edge Computing. Summarized MEC Models. Classification of Resource Management.
	[198]	Workload allocation estimation between fog and cloud. Minimum power consumption versus service delays modeling.
	[199]	Device-driven and human-driven ML based intelligence schemes. Cross-layers optimization involving efficient MAC layer scheduling and fog data offloading.
Access Networks	[186]	System architecture for F-Radio Access Networks (RANs). Edge caching, software-defined networking and network-function virtualization.
	[200]	Model design of cache management in enhanced remote radios
Networks: Management, Virtualization & Orchestration	[201]	Compute enabled Fog Nodes. Process and resources isolation using virtual machine Fog Node architecture. Inter and Intra Fog Nodes communication, VM migration and traffic minimization by software-defined core.
	[202]	Models a Fog orchestration scenario for network functions.
	[203]	Virtual Fog framework to support Object and Network virtualization.
Security & Privacy	[204]	The proposed model to revoke security certificates for improved privacy and security in IoT Networks.
	[205]	Models a security attack on a Fog device.
	[206]	Security threats and solutions overview for Fog and IoT applications.

## Data Availability

No new data were created or analyzed in this study. Data sharing is not applicable to this article.

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
