# Peer review of "A Comprehensive Review of Internet of Things: Technology Stack, Middlewares, and Fog/Edge Computing Interface"

_sensors, 2022, doi:10.3390/s22030995_

Round 1

Reviewer 1 Report

This research article aims to provide a comprehensive overview of the enabling technologies and standards that builds up the IoT technology stack. First, a layered architecture approach is presented where the state-of-the-art research and open challenges are discussed at every layer. Next, this research article focuses on the role of middleware platforms in IoT application development and integration. Furthermore, this article addresses the open challenges and provides comprehensive steps towards IoT stack optimization. Finally, the interfacing of Fog/Edge Networks to IoT technology stack is thoroughly
investigated by the discussing the current research and open challenges in this domain. 
1)It is better to give an application scenario for the considered systems.
2)The presentation style of this work can be improved. In particular, some figures such as Figs. 1-2 are presented in a poor style.
3)It is better for the authors to discuss the implementation issue of the proposed system in practice.
4)There are some typos in the paper, which should be corrected carefully.
5)Some recent works on the IoT networks should be incorporated into the paper, in order to enhance the literature review, such as
[1] W. Zhou, "PSO Based Offloading Strategy for Cache-Enabled Mobile Edge Computing UAV Networks," Cluster Computing, vol. 2021, no. 24, pp. 1-13, 2021;
6)Please give some more discussion on the future works in the part of conclusions.
7)It is better to give some more discussions on the comparisons among different methods.
8)The authors should list the main contributions of this work, at the end of Introduction.

Reviewer 2 Report

This paper is a review of internet of things. The state-of-the art research and open challenges in IoT are discussed in this paper. Three aspects in IoT are presented, including technology stack, middlewares and fog/edge computing interface. However, I think that the main content of this paper is not relevant to IoT. The drawbacks of this paper are listed as follows.

1. A review article should focus on the studies of technologies, protocols and algorithms. The organization of this paper seems like a report rather than a research article. For example, the authors present the search terms on page 5. The analysis of the business opportunities is too detailed on page 6.

2. The paper lacks technical clarity and is full of confusing sentences and grammatical errors. Moreover, some figures in this paper are not suitable for publishing on the research paper, such as the figures of 9 and 11.

3. There is no technical difference in the studies comparison, and only the contribution comparision cannot reflect the value of the reviewed articles. The analysis of the challenges and problems are not specific enough.

Hence, the paper is not suitable for this journal, since the work of this paper has little relationship with the topic. So I suggest the authors revising the paper seriously and deliver the paper to another suitable journal or conference.

Reviewer 3 Report

This paper presents a survey on IoT and edge and fog computing. There are a good collection of information is presented in this paper which might be interesting for other researchers. 

The main issue in this paper is that it merged two surveys into one. The first one is about IoT and the second is about Fog/Edge computing. That's why the paper is so long.  The paper doesn't have a focus and it confuses the reader middle of the paper. 

I would suggest the authors modify the content and present a more focused survey that includes better insights for readers. 

Moreover, the paper doesn't finish with strong discussions or conclusions. Section 8 should be revised with more insightful discussions and some approaches for future works.

There are many tables in the papers and they are not consistent and they have different formats. 

Finally, it would be good to have a motivation statement and comparison to other survey papers to distinguish this survey and to understand why we need to have this survey.  

Round 2

Reviewer 1 Report

The paper has been well revised, and the current version is ready for publication.

Reviewer 2 Report

The author has made a lot of modifications and improvements in the revision. As a  review paper,  the organization and content have met the requirements of a survey. It is hoped that the author should carefully check the sentences and further correct the typos in this paper.

Reviewer 3 Report

The authors addressed my comments.
